# Efficient Equivariant Transfer Learning from Pretrained Models

**Sourya Basu** [*]
University of Illinois at Urbana-Champaign

**Pulkit Katdare** [*]
University of Illinois at Urbana-Champaign

**Prasanna Sattigeri**
IBM Research

**Vijil Chenthamarakshan**
IBM Research

**Katherine Driggs-Campbell**
University of Illinois at Urbana-Champaign

**Payel Das**
IBM Research

**Lav R. Varshney**
University of Illinois at Urbana-Champaign

## Abstract

Efficient transfer learning algorithms are key to the success of foundation models on diverse downstream tasks even with limited data. Recent works of Basu et al. (2023) and Kaba et al. (2022) propose group averaging (*equitune*) and optimization-based methods, respectively, over features from group-transformed inputs to obtain equivariant outputs from non-equivariant neural networks. While Kaba et al. (2022) are only concerned with training from scratch, we find that equitune performs poorly on equivariant zero-shot tasks despite good finetuning results. We hypothesize that this is because pretrained models provide better quality features for certain transformations than others and simply averaging them is deleterious. Hence, we propose $\lambda$-*equitune* that averages the features using *importance weights*, $\lambda$s. These weights are learned directly from the data using a small neural network, leading to excellent zero-shot and finetuned results that outperform equitune. Further, we prove that $\lambda$-equitune is equivariant and a universal approximator of equivariant functions. Additionally, we show that the method of Kaba et al. (2022) used with appropriate loss functions, which we call *equizero*, also gives excellent zero-shot and finetuned performance. Both equitune and equizero are special cases of $\lambda$-equitune. To show the simplicity and generality of our method, we validate on a wide range of diverse applications and models such as 1) image classification using CLIP, 2) deep Q-learning, 3) fairness in natural language generation (NLG), 4) compositional generalization in languages, and 5) image classification using pretrained CNNs such as Resnet and Alexnet.

## 1 Introduction

Group-equivariant deep learning leverages group equivariance as an inductive bias to design efficient and reliable neural networks. Popular examples include convolutional neural networks (CNNs) equivariant to translations (LeCun et al., 1989), group convolutional neural networks (GCNNs) equivariant to general discrete groups (Cohen & Welling, 2016), and recently Alphafold2 equivariant to 3D rotations (Jumper et al., 2021). But these methods cannot leverage pretrained models.

With the increase in open sourced large pretrained models, it is now crucial to develop efficient transfer learning algorithms that can leverage group equivariance. Basu et al. (2023) proposed equitune, an equivariant finetuning method that uses group averaging over features extracted from pretrained

---

[*]Equal contribution.

37th Conference on Neural Information Processing Systems (NeurIPS 2023).

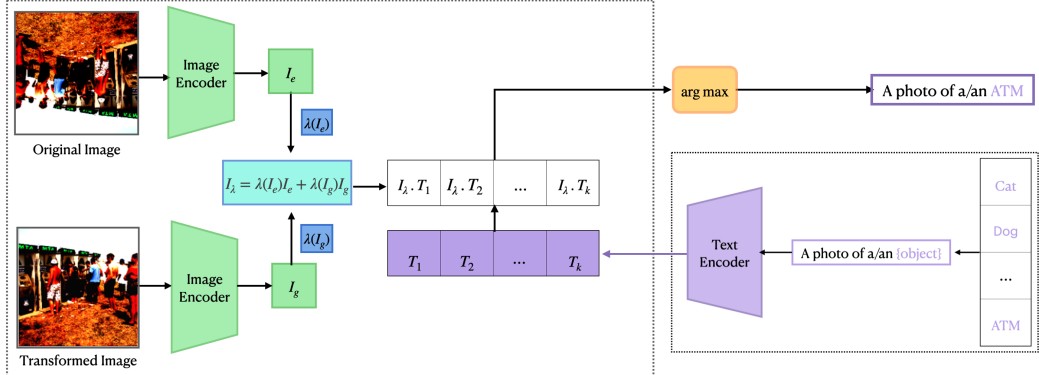

Figure 1: Implementation of $\lambda$-equitune on CLIP. Weighted average of image features corresponding to transformed inputs are computed, which is further used for computing text-image similarity scores.

models. Several other methods were proposed to get equivariant output from non-equivariant backbone architectures, e.g. some form of averaging (Puny et al., 2021; Atzmon et al., 2022) or optimization over certain proxy loss functions (Kaba et al., 2022). But these latter methods were originally designed for training from scratch and not much is known about their finetuning abilities.

Here, we find that equitune performs poorly on zero-shot tasks. Our first main contribution is to show that the optimization method of Kaba et al. (2022) when used with appropriate proxy loss functions provides excellent zero-shot and finetuning performance. We call this method *equizero*.

The results from equizero suggest that pretrained models provide better quality features for some group transformations than others. Thus, we propose $\lambda$-equitune, which learns *importance weights* directly from the data and uses them to perform a weighted group averaging. We show $\lambda$-equitune outperforms equitune and is competitive with equizero for zero-shot tasks. Moreover, for finetuning, $\lambda$-equitune often outperforms both equitune and equizero. This constitutes our second main contribution.

To validate our methods, we provide experiments on a diverse set of pretrained models and datasets. We show zero-shot performance of equizero on: 1) image classification using CLIP, 2) deep Q-learning, 3) fairness in natural language generation (NLG), and 4) compositional generalization in languages. For CLIP and deep Q-learning, we used the naturally available loss functions, namely, similarity scores and $Q$-values, respectively. For fairness in NLG and compositional generalization, we closely follow the setup of Basu et al. (2023), and we use *regard scores* Sheng et al. (2019) and the negative of the maximum of the output probability distribution, respectively, as the loss function.

We first show results of $\lambda$-equitune for image classification using CLIP, finding that $\lambda$-equitune performs competitively with equizero and outperforms equitune for both zero-shot and finetuning tasks. Then, we show a simple case where finding a good proxy loss function for equizero is non-trivial: image classification using pretrained CNNs. Here, we find that equizero performs even worse than equitune but $\lambda$-equitune easily outperforms equitune and equizero. The organization of the paper is summarized below:

- §3 provides details of $\lambda$-equitune and equizero and proves a number of their properties.

- §4 provides overview of the applications considered in the experiments and how equivariance methods are used there.

- §5 provides experimental details and results for all the applications.

## 2   Background

Here we discuss relevant concepts in group equivariance and group equivariant transfer learning.

**Group Equivariance**   A **group** $(G, \cdot)$ is a set $G$ accompanied by a binary operator '$\cdot$' that satisfy the axioms of a group, namely i) closure: $g \cdot h \in G$ for all $g, h \in G$; ii) associativity: $(g \cdot h) \cdot k = g \cdot (h \cdot k)$;

iii) identity: there exists $e \in G$, such that $g \cdot e = e \cdot g$ for all $g \in G$; iv) inverse: for every element $g \in G$, there exists $g^{-1}$, such that $g \cdot g^{-1} = g^{-1} \cdot g = e$. We write $g \cdot h$ as $gh$ for brevity.

Given a set $\mathcal{X}$, we define a **group action** of $G$ on $\mathcal{X}$ as $\Gamma_{\mathcal{X}} : G \times \mathcal{X} \mapsto \mathcal{X}$ such that it satisfies two axioms, namely i) identity: $\Gamma_{\mathcal{X}}(e, x) = x$ for all $x \in \mathcal{X}$, where $e \in G$ is the identity; ii) compatibility: $\Gamma_{\mathcal{X}}(g, \Gamma_{\mathcal{X}}(h, x)) = \Gamma_{\mathcal{X}}(gh, x)$, for all $g, h \in G, x \in \mathcal{X}$. We write $\Gamma_{\mathcal{X}}(g, x)$ simply as $gx$ for brevity.

A model $\mathbf{M} : \mathcal{X} \mapsto \mathcal{Y}$ is **equivariant** to $G$ under the group action of $G$ on $\mathcal{X}$ and $\mathcal{Y}$ if $\mathbf{M}(gx) = g\mathbf{M}(x))$ for all $g \in G, x \in \mathcal{X}$. This essentially means that any group transformation $g$ to the input $\Gamma_{\mathcal{X}}(g, x)$ should reflect with an equivalent group transformation of the output $\Gamma_{\mathcal{Y}}(g, \mathbf{M}(x))$.

**Equivariant Finetuning** Recently, Basu et al. (2023) proposed a finetuning method called equituning that starts with potentially non-equivariant model $\mathbf{M}$ and produces a model $\mathbf{M_G}$ that is equivariant to $G$. Equituning converts a pretrained model into an equivariant version by minimizing the distance of features obtained from pretrained and equivariant models. The output of an equituned model is given by

$$\mathbf{M_G}(x) = \frac{1}{|G|} \sum_{g \in G} g^{-1} \mathbf{M}(gx). \tag{1}$$

While the averaging in equation 1 is shown to be useful for finetuning, we find it leads to poor equivariant zero-shot learning. This could be because the pretrained model outputs high quality features only for some of the transformed inputs. Hence, averaging them directly leads to low quality zero-shot performance. This can be avoided by using weighted averaging as discussed in Sec. 3.

**Optimization-Based Canonical Representation** On the other hand, Kaba et al. (2022) show that group equivariant output, $\mathbf{M_G^{OPT}}(x)$, can be obtained by optimizing a (non-equivariant) loss function $l(\mathbf{M}(gx))$ with respect to group elements $g \in G$ for any $x \in \mathcal{X}$ as shown below.

$$\mathbf{M_G^{OPT}}(x) = g_*^{-1} \mathbf{M}(g_* x), \tag{2}$$

where $g_* = \arg\min_{g \in G} l(\mathbf{M}(gx))$ with $l : \mathcal{Y} \mapsto \mathbb{R}$ being an injective proxy loss function and assuming the minima is unique. However, the purpose of this formulation in Kaba et al. (2022) is only to obtain an equivariant representation for training from scratch. Moreover, no competitive zero-shot or finetuning performance is obtained in Kaba et al. (2022) using this method. We show that the choice of $l$ plays an important role in its zero-shot and finetuning performance, even outperforming equituning. This method is obtained as a special case of $\lambda$-equituning introduced next.

**Additional Related Works** Group equivariance plays a key role in geometric deep learning (Bronstein et al., 2021) for designing efficient domain specific neural networks. Several elegant architectures have been proposed for equivariant image classification (Cohen & Welling, 2016, 2017; Romero & Cordonnier, 2020), reinforcement learning (Mondal et al., 2020; van der Pol et al., 2020; Mondal et al., 2022; Wang et al., 2022), graph (Satorras et al., 2021; Keriven & Peyré, 2019; Gasteiger et al., 2021) and mesh (De Haan et al., 2020; He et al., 2021; Basu et al., 2022) processing, natural language processing (Gordon et al., 2019; Li et al., 2022), and data generation (Dey et al., 2021). These architectures need to be trained from scratch, which is not always desirable.

*Frame* averaging produces equivariant output from non-equivariant architecture backbones (Puny et al., 2021; Atzmon et al., 2022; Duval et al., 2023). Most work here focuses on finding good frames, which are equivariant subsets of groups, for specific groups, and not for general groups. And not much is known about their performance with pretrained models. Kaba et al. (2022) also give a canonicalization-based method that uses an equivariant auxiliary network for constructing equivariant networks out of non-equivariant backbones and is used for training from scratch. But this work requires an additional equivariant network and appropriate trainable parameterization of the group actions, which is presented only for certain groups of interest. Further, this work is not concerned with equivariant performance of pretrained models. Zero-shot group equivariance was also recently used by Muglich et al. (2022) for zero-shot coordination in partially observable Markov decision processes (POMDPs). In contrast, our work aims to be provide efficient equivariant transfer learning algorithms that are general in terms of considered tasks and groups, and does not require additional equivariant networks.

# 3  $\lambda$-Equitune

We propose $\lambda$-equitune, where unequal weights are assigned to features obtained from transformed inputs. This is a simple generalization of equitune in equation 1 and the optimization-based approach in equation 2, where the goal is to assign higher values to better features. Like these previous methods, $\lambda$-equitune is equivariant and a universal approximator of equivariant functions.

The main idea of $\lambda$-equitune is that given a pretrained model $\mathbf{M}$, the features $\mathbf{M}(gx)$ for any fixed $x$ are not all equally important for all $g \in G$. We denote by $\lambda(gx)$ the *importance weight* of feature $\mathbf{M}(gx)$ for $g \in G, x \in \mathcal{X}$. We assume $G$ is finite, just as in Basu et al. (2023). Suppose $\lambda : \mathcal{X} \mapsto \mathbb{R}^+$ is known a priori, and denote the $\lambda$-equituned model as $\mathbf{M}_{\mathbf{G}}^{\lambda}$. Then we want to minimize

$$\min_{\mathbf{M}_{\mathbf{G}}^{\lambda}(x)} \quad \sum_{g \in G} \left\| \lambda(gx)\mathbf{M}(gx) - \mathbf{M}_{\mathbf{G}}^{\lambda}(g, x) \right\|_2^2$$
$$\text{s.t.} \quad \mathbf{M}_{\mathbf{G}}^{\lambda}(gx) = g\mathbf{M}_{\mathbf{G}}^{\lambda}(x) \text{ for all } g \in G. \tag{3}$$

We call the solution to equation 3 as $\lambda$-equitune, given by

$$\mathbf{M}_{\mathbf{G}}^{\lambda}(x) = \frac{1}{\sum_{g \in G} \lambda(gx)} \sum_{g \in G}^{|G|} g^{-1}\lambda(gx)\mathbf{M}(gx). \tag{4}$$

## 3.1  Special Cases

When $\lambda(gx) = 1$ for $g \in G$, then equation 4 is the same as equation 1. Further, we get equation 2 when $\lambda$ is an indicator function $\lambda(gx) = \mathbb{1}_{\{g=g_*\}}$, where $g_* = \arg\min_{g \in G} l(\mathbf{M}(gx))$ with $l : \mathcal{Y} \mapsto \mathbb{R}$ such that the minimization is well defined. We use $\mathbf{M}_{\mathbf{G}}^{\mathbf{0}}$ to denote the equizero model.

The first main contribution of our work is experimental. We show there are good loss functions for equizero for several diverse tasks that easily outperform equitune by choosing the best features.

The second main contribution is to show that $\lambda$-equitune outperforms equitune and is competitive with equizero even when good loss functions are known. Moreover, when the loss functions are not trivial, equizero performs even worse than equitune, but $\lambda$-equitune easily outperforms both.

## 3.2  Canonical-$\lambda$-Equitune

Here, we provide an extension of the $\lambda$-equitune algorithm of equation 4 to continuous groups. The idea is to combine the canonicalization method of Kaba et al. (2022) with $\lambda$-equitune leading to an expressive equivariant network with weighted averaging over features with different group actions applied to them.

**Definition 1** (Canonical-$\lambda$-equitune). *Given a (continuous) group $G$, a non-equivariant function $M : X \mapsto Y$, and an equivariant auxiliary function (from the setting of Kaba et al. (2022)) $h : X \mapsto G$, lambda functions $\lambda : X \mapsto R^+$, and a set of group elements $\Theta = \{\theta_1, \ldots, \theta_k\}$, i.e. $\theta_i \in G$, we define the canonical-$\lambda$-equitune operators as*

$$M_{G,equi}^{\lambda}(x) = \frac{1}{\sum_{\theta \in \Theta} \lambda(\theta h(x)^{-1}x)} \sum_{\theta \in \Theta} \lambda(\theta h(x)^{-1}x)h(x)M(\theta h(x)^{-1}x) \tag{5}$$

$$M_{G,inv}^{\lambda}(x) = \frac{1}{\sum_{\theta \in \Theta} \lambda(\theta h(x)^{-1}x)} \sum_{\theta \in \Theta} \lambda(\theta h(x)^{-1}x)M(\theta h(x)^{-1}x). \tag{6}$$

In Thm. 1, we show that the canonical-$\lambda$-equivariant network is equivariant to the group $G$

**Theorem 1.** $M_{G,equi}^{\lambda}(x)$ *and* $M_{G,inv}^{\lambda}(x)$ *are, respectively, equivariant and invariant to $G$.*

## 3.3  Properties

Now we show in Thm. 2 that equation 4 is equivariant with respect to $G$.

**Theorem 2** (Equivariance). $\mathbf{M}_{\mathbf{G}}^{\lambda}$ *defined in equation 4 is equivariant to $G$.*

Table 1: Inference times of equitune, equizero, and $\lambda$-equitune for the c4 group for various CLIP models on CIFAR100. We use batch size 32 on a single Nvidia A100 GPU.

| Model | Time (sec.) | | | Memory (MB) | | |
|---|---|---|---|---|---|---|
| | Equitune | Equizero | $\lambda$-Equitune | Equitune | Equizero | $\lambda$-Equitune |
| RN50 | 14.15 | 14.10 | 23.5 | 3703 | 3703 | 3941 |
| ViT-B/32 | 11.00 | 10.27 | 16.08 | 2589 | 2587 | 2915 |

We define a universal approximator in Def. 2. Then, Thm. 3 proves that $\lambda$-equitune is a universal approximator of equivariant functions for groups where $\|g\| = 1$ for $g \in G$. This includes a wide range of groups including the permutation group, the $SO(n)$ groups of special orthogonal groups, etc. This condition is the same as the class of groups considered in Basu et al. (2023).

**Definition 2** (Universal approximator). *A model $\mathbf{M} : \mathcal{X} \mapsto \mathcal{Y}$ is a universal approximator of a continuous function $f : \mathcal{X} \mapsto \mathcal{Y}$ if for any compact set $\mathcal{K} \subset \mathcal{X}$ and $\epsilon > 0$, there exists a choice of parameters for $\mathbf{M}$ such that $\|f(x) - \mathbf{M}(x)\| \le \epsilon$ for all $x \in \mathcal{K}$.*

**Theorem 3** (Universality). *Let $f_G : \mathcal{X} \mapsto \mathcal{Y}$ be any continuous function equivariant to group $G$ and let $\lambda : \mathcal{X} \mapsto \mathbb{R}^+$ be any positive scalar function. And let $\mathbf{M} : \mathcal{X} \mapsto \mathcal{Y}$ be a universal approximator of $\frac{f_G}{\lambda}$. Here $\mathcal{X}$ is such that if $x \in \mathcal{X}$, then $gx \in \mathcal{X}$ to ensure the equivariance of $f_G$ is well-defined. Then, $\mathbf{M}_{\mathbf{G}}^\lambda$ is a universal approximator of $f_G$.*

**Computational Complexity** Note that equitune, equizero, and $\lambda$-equitune have the same compute complexity. In practice, in Tab. 1 we find that equitune and equizero have exactly the same time and memory consumption, whereas $\lambda$-equitune takes a little more time and memory because of the additional $\lambda$ network. We illustrate on RN50 and ViT-B/32 models of CLIP using the same text encoder, but different image encoders. RN50 uses a Resnet50 based image encoder, whereas ViT-B/32 uses a vision transformer based image encoder.

**Beyond Zero-Shot Learning** Let us emphasize that even though the equizero model in equation 2 is not differentiable due to the $\arg\max$, we can still use simple gradient estimators known in the literature. One popular estimator is the straight-through estimator (Bengio et al., 2013), where the equizero output in equation 2 would be written as $\mathbf{M}_{\mathbf{G}}^{\mathbf{0}}(x) = \mathbf{M}(x) + (\mathbf{M}_{\mathbf{G}}^{\mathbf{0}}(x) - \mathbf{M}(x)).\texttt{detach()}$, where $\texttt{detach()}$ indicates that no gradient flows through the term $(\mathbf{M}_{\mathbf{G}}^{\mathbf{0}}(x) - \mathbf{M}(x))$. In practice, we found it to be slightly better to use $\mathbf{M}_{\mathbf{G}}(x)$ instead of $\mathbf{M}(x)$ and write $\mathbf{M}_{\mathbf{G}}^{\mathbf{0}}(x) = \mathbf{M}_{\mathbf{G}}(x) + (\mathbf{M}_{\mathbf{G}}^{\mathbf{0}}(x) - \mathbf{M}_{\mathbf{G}}(x)).\texttt{detach()}$. §5.1.3 illustrates few-shot learning and finetuning using equizero and compares with equituning.

## 4 Applications

First we present several applications in §4.1 where finding a loss function for equizero is easy. This naturally leads to excellent equivariant zero-shot results outperforming equitune. Then, in §4.2 we provide two applications in image classification to show 1) the benefits and drawbacks of equizero compared to equitune and $\lambda$-equitune, and 2) that $\lambda$-equitune consistently performs well avoiding the drawbacks of equizero.

### 4.1 Equizero Applications

Here we provide applications where equizero achieves excellent zero-shot performance, namely: 1) deep Q-learning, 2) fairness in NLG, and 3) compositional generalization in languages.

**Equizero Reinforcement Learning** Recent works, such as van der Pol et al. (2020); Mondal et al. (2020), have developed RL algorithms that leverage symmetries in the environments that helps improve robustness and sample efficiency. But no existing work efficiently uses group equivariance on top of pretrained RL models. Here we apply equizero and equitune on top of pretrained models inspired from the group symmetries found in van der Pol et al. (2020). We find that equitune outperforms non-equivariant pretrained models but equizero outperforms both. We simply use the $Q$-values as the proxy loss function as described in §C.1 with more details.

**Group-Theoretic Fairness in NLG**   We seek to reduce the social biases inherent in language models (LMs), focusing on GPT2 (Radford et al., 2019). We consider the group-theoretic fairness setup of Sheng et al. (2019) and Basu et al. (2023). We take the sets of demographics, namely ['man', 'woman'], ['straight', 'gay'], and ['black', 'white']. For each demographic group, Sheng et al. (2019) proposed two tasks, called *respect task* and *occupation task*, where each task consists of five context phrases. A language model (LM) is given these contexts to generate sentences. These generated sentences are then classified as 'positive', 'negative', 'neutral', or 'other' by a *regard classifier* also proposed by Sheng et al. (2019). These outputs are called regard scores. A regard classifier is a BERT-based model similar to a sentiment classifier but more specific for fairness tasks. We use equizero using the regard scores as the proxy loss function, so as to maximize positivity in the generated texts, while guaranteeing group-theoretic fairness. To that end, we propose two algorithms that help in reducing existing biases across demographic groups.

*EquizeroLM and R-EquizeroLM:* Basu et al. (2023) define EquiLM and R-EquiLM, that use a sequential version of equitune to perform group transformed averaging to achieve fairness across demographic groups. While EquiLM and R-EquiLM generate debiased outputs, they do not produce positive regard scores, which is desirable to reduce toxicity in generated text. We propose EquizeroLM and R-EquizeroLM which use equizero to maximize regard scores, ensuring both fair and less toxic. Further details provided in §C.2.

**Zero-Shot Compositional Generalization**   We show compositional generalization capabilities of equizero on the SCAN dataset (Lake & Baroni, 2018). SCAN measures compositionality using a language-to-action translation task. E.g., if the model learns that the phrase "Jump", "Run", "Run Twice" translate to the actions "JUMP", "RUN", "RUN RUN" from the train set, then, SCAN tests whether the model also learns that "Jump Twice" translates to "JUMP JUMP". Such a reasoning however common in human beings is hard to find in language models.

We apply equizero on two tasks in SCAN, *Add Jump* and *Around Right*. Gordon et al. (2019) solved these tasks by constructing sophisticated group equivariant networks from scratch and training them. Basu et al. (2023) used the same group actions as Gordon et al. (2019), but used equituning on pretrained non-equivariant models for a few iterations and obtained comparable results. But, as we note, equitune has poor zero-shot performance. We show that equizero using negative of maximum probability from the output as the loss function gives much better zero-shot performance. Using gradient estimators described in §3.3, we also compare the finetuning performance of equizero against equitune. Details of group actions used are given in §C.3

## 4.2   $\lambda$-Equitune Applications

Here we consider two important applications: CLIP-based and CNN-based image classification. For CLIP, it is easy to find a loss function for equizero that provides better results than equitune. But for the simple case of CNN-based classification it is non-trivial to find such a loss function. Since $\lambda$-equitune does not require a loss function, it performs better than equitune in both cases. Equizero only performs well for CLIP, but fails miserably for CNN-based classification.

**CLIP-Based Image Classification**   CLIP is a pretrained model consisting of image and text encoders that give impressive zero-shot classification performance across variety of unseen datasets. But, in Fig. 4a and 7, we find that CLIP is not robust to simple transformations such as rotation by multiples of $90°$ or random flips. This trend is seen across different image encoders like RN50, RN101 (Radford et al., 2021), ViT-B/32 and ViT-B/16 (Dosovitskiy et al., 2021). This can be avoided by making the model in/equi-variant to such transformations, e.g., by using equitune. But we show in §5.2.1 that equitune does not produce good zero-shot performance. We show in §5.2.1 that using equizero with image-text similarity score as loss function provides much better zero-shot results than equitune. Later, in §5.2.1, we show that $\lambda$-equitune achieves better zero-shot performance than equitune without any loss function. Finally, when finetuned, we find that $\lambda$-equitune tends to perform the best, possibly because it does not need gradient estimators like equizero, and because it uses weighted averaging to obtain better features than equitune.

**CNN-based image classification**   For image classification using pretrained CNNs such as Resnet and Alexnet, we note that finding a good loss function for equizero is non-trivial. As such, we consider two loss functions 1) negative of the maximum probability as it worked well with the SCAN

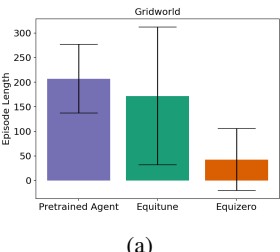
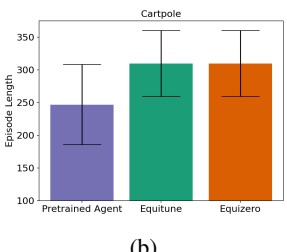
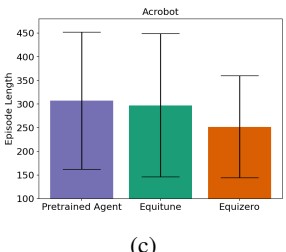

|  (a)  |  (b)  |  (c)  |

Figure 2: Comparison of zero-shot performance of equizero to equituning and a non-equivaraint pretrained model are shown in (a), (b), and (c) for Gridworld, Cartpole, and Acrobot, respectively. Equizero outperforms both equituning and non-equivariant pretrained model. Results over five seeds.

task in §4.1 and 2) the entropy of the output distribution since it correlates with the confidence of the model (Wang et al., 2018). But, equizero with these loss functions performs even worse than equitune. Further, we find that $\lambda$-equitune easily outperforms both equitune and equizero.

# 5 Experiments

Here, we provide experimental results for equizero and $\lambda$-equitune in §5.1 and §5.2, respectively, for all the applications described in §4. Additional experiments for canonical-$\lambda$-equitune are provided in §. D.5. The code for this paper is available at `https://github.com/basusourya/lambda_equitune`.

## 5.1 Zero-Shot Performance using Equizero

### 5.1.1 Equizero Reinforcement Learning

**Experimental Setting:** We first pretrain Deep Q-learning nets (DQNs) for each of the Gridworld, Cartpole, and Acrobot environments using the default architecture from Raffin et al. (2021) with 103k parameters. We pretrained all the models using a learning rate $10^{-4}$. We used training time steps as 100k, 100k, and 70k for Gridworld, Cartpole, and Acrobot, respectively. These number of steps were chosen to obtain the best models by varying the time steps from 50k to 100k in multiple of 10k for a fixed seed.

**Results and Observations:** Fig. 2 show the evaluation performance of equizero and compare it with equituning and non-equivariant pretrained models. We find that equituning performs better than non-equivariant models and equizero outperform both of them. The results are over five seeds.

### 5.1.2 Fairness in Natural Language Generation

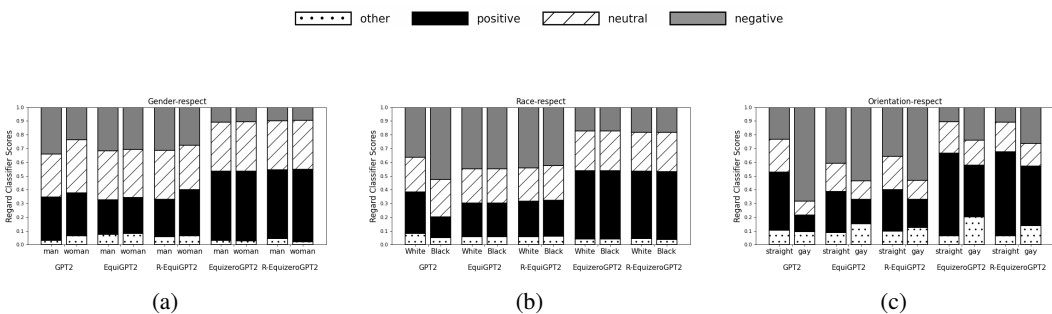

|  (a)  |  (b)  |  (c)  |

Figure 3: Plots (a), (b), and (c) show the regard scores for GPT2, EquiGPT2, R-EquiGPT2, EquizeroGPT2, and R-EquizeroGPT2. In equitune, if negativity is present in some demographics, it gets redistributed in the other demographics, which is undesirable. Equitune is only able to debias, whereas, equizero models not only debiases the texts but also makes the regard scores more positive.

Table 2: Zero-shot performance of non-equivariant models, equituned, and equizeroed models for LSTM on SCAN. LSTMs were trained for 200K iterations. We find that equizero outperforms other methods using non-equivariant pretrained models. Results are over three random seeds.

| | Add Jump | | | Around Right | | |
| Model | Group | Val. Acc. | Test Acc. | Group | Val. Acc. | Test Acc. |
|---|---|---|---|---|---|---|
| LSTM | – | 99.1 (0.3) | 0.0 (0.0) | – | 98.9 (0.7) | 0.4 (0.7) |
| EquiLSTM | Verb | 62.5 (1.5) | 7.1 (1.3) | Direction | 80.5 (4.5) | 28.2 (12.9) |
| EquizeroLSTM | Verb | 98.4 (0.9) | **75.2 (1.5)** | Direction | 98.7 (5.8) | **81.7 (2.4)** |

**Experimental Setting**: We use GPT2 (Radford et al., 2019), with 117M parameters as our pretrained model. We consider the lists of demographic groups ['man', 'woman'], ['white', 'black'], and ['straight', 'gay' ]. We compare our method against EquiGPT2 and R-EquiGPT2 (Basu et al., 2023). For the equizero models, we use beam length $m$ as 5, as described in §C.2. We limit the sentence lengths to be 15 for all models. We generated 500 sentences for each of the models and demographic groups by varying the seeds for both respect and occupation context.

**Results and Observations**: Fig. 3 and 9 compare the regard scores for all the considered models. We observe that equizero models are not only able to debias among various demographics like 'man' and 'woman', but it also reduces the toxicity/ negativity of the scores. Debiasing is seen from the equality of the scores amongst the demographics considered for equivariance. And reduction in toxicity is observed by noticing that the regard scores are more positive. Like equituning (Basu et al., 2023), equizero models show high quality of generated texts. Sample generations from the demographic groups ['straight', 'gay'] are shown in Tab. 3, 6, 7, and 5 for all the models. Note in Tab. 3 and 6, that even with perfect equivariance, where the word 'straight' simply gets replaced by 'gay', the regard scores are very different. This shows the presence of bias in the regard classifier itself as was also observed by Basu et al. (2023).

### 5.1.3 Compositional Generalization using Equizero

**Experimental Setting:** We evaluate the performance of our algorithm on the SCAN Dataset (Gordon et al., 2019) on the *Add Jump* and *Around Right* tasks. All the recurrent models (RNN, LSTM, GRU) and their equivariant counterparts contain a single hidden layer of 64 units. For all training processes, we use the Adam optimizer (Kingma & Ba, 2015) and teacher-forcing ratio 0.5 (Williams & Zipser, 1989). For pretraining and finetuning, we use 200k and 10k iterations respectively. For pretraining, we use a learning rate of $10^{-4}$, whereas for finetuning, we used learning rates $10^{-5}$ and $2 \times 10^{-5}$ for *Add Jump* and *Around Right*, respectively. Results are over three seeds.

**Observation and Results:** Tab. 2 shows that equizero outperforms equituned and non-equivariant models on zero-shot performance with LSTMs. In §D, we observe similar results for RNNs and GRUs in Tab. 9 and 8, respectively. We use gradient estimators discussed in §3.3 for performing few-shot learning using equizero. For few-shot learning, we find in Fig. 10 in §D that equizero is competitive with equitune for small iterations, but as the number of steps increase, equitune is the better algorithm. This is expected since the gradients are computable for equitune, but only approximated in equizero. Tab. 10, 11, and 12 in §D provide the results for finetuning using equitune and equizero for 10k iterations. We find the equitune is slightly better when finetuning for 10k iterations. This shows equizero is better for zero-shot and few-shot learning, but for large iterations, equitune is preferable.

## 5.2 Zero-Shot and Finetuning Performance using $\lambda$-Equitune

### 5.2.1 Equi/Invariant Image Classification using CLIP

**Experimental Setting:** We first perform zero-shot image classification on Imagenet-V2 and CI-FAR100 using CLIP. We use two transforms, random $90°$ rotations and flips, for testing their robustness. We encode class labels using the 80 text templates provided in Radford et al. (2021) [2].

---

[2] Obtained from `https://github.com/openai/CLIP`

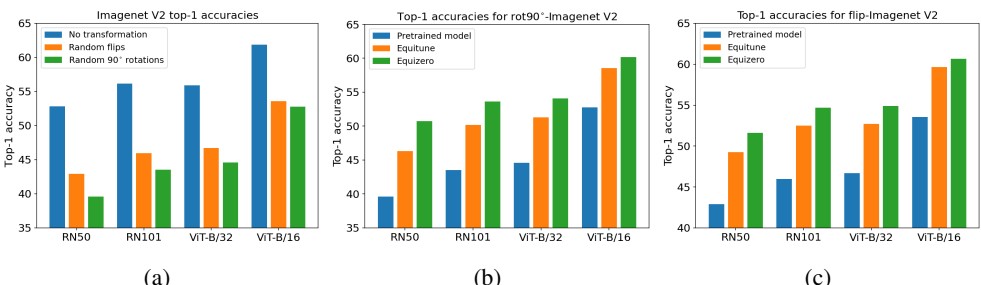

(a)          (b)          (c)

Figure 4: In (a) note that zero-shot performance of CLIP drops significantly when we add random rotations or flips to the input. This trend is seen across all image encoders, i.e. ResNet (RN50 and RN101) and ViT (ViT-B/32 and ViT-B/16). In (b) and (c) we show classification results on Imagenet-V2 with random $90°$ rotations and flips, respectively. We observe equizero outperform equitune and original CLIP for all image encoders.

We then evaluate finetuning capabilities of equitune, equizero, and $\lambda$-equitune on CIFAR100 with random $90°$ rotations. Here we choose $\lambda$ to be a two-layered feed-forward network, which is described in detail along with learning rates used, in §D.4. The input to this $\lambda$-network are the features from a frozen CLIP encoder. First, the $\lambda$-network is trained with the model kept frozen. Then, while finetuning the model, the $\lambda$ network is frozen. We show results for both Resnet and ViT backbone trained for 1000, 2000, 3000, 4000 finetuning steps.

**Results and Observations:** Fig. 4 shows test accuracies for Imagenet-V2 with random $90°$ rotations and flips. We observe that the pretrained model's performance reduces drastically when transformations are applied to the dataset. Whereas both equitune and equizero are relatively robust to such transformations. Moreover, equizero outperforms both equitune and the pretrained model. Similar observations are made for CIFAR100 in Fig. 8 in §D.

In Fig. 5a and 11a we plot the test accuracies of $\lambda$-equitune on CIFAR100 for both variants of Resnet and ViT backbones. We observe that $\lambda$-equitune performs better than both equitune and equizero (with finetuning) on Resnets. On ViT-B/16, we observe that $\lambda$-equitune easily outperforms both equitune and equizero (with finetuning). On ViT-B/32, we find that $\lambda$-equitune outperforms equitune but equizero outperforms both equitune and $\lambda$-equitune. Thus, $\lambda$-equitune performs competitively with equizero, even in applications where good loss functions are known.

### 5.2.2 Equi/Invariant Image Classification using Pretrained CNNs

**Experimental Setting:** We now evaluate the performance of $\lambda$-equitune on rot90-CIFAR10 dataset. Here, each image is rotated randomly by a multiple of $90°$. We use pretrained Alexnet and Resnet, trained extensively over CIFAR100. For the $\lambda$-equituning, we choose $\lambda$ as a two layered feed-forward network with a hidden layer of dimension 100 with input as features extracted by this pretrained Alexnet or Resnet. Along with $\lambda$, we also perform linear probing wherein last two layers for the classification problem is being learned using a learning rate of $10^{-3}$.

**Observation and Results:** In Fig. 5b and 11b we see that $\lambda$-equitune outperforms equitune and equizero. Moreover, equizero performs even worse than equitune. This trend is consistent across both Alexnet and Resnet pretrained modules.

## 6 Limitations, Societal Impact, and Conclusion

**Limitations and Societal Impact:** Our results focus on finite groups; extension to continuous groups requires further work in parameterization of continuous groups (cf. Benton et al. (2020)) or group decomposition (cf. Basu et al. (2021); Maile et al. (2023)). Our work on fairness in NLG aims to debias foundation models and thereby lead to positive societal impact. But we use equality and neutral sets obtained from previous human-made works. We believe there is scope for optimizing the design of such sets using neural networks for more general demographic groups.

**Conclusion:** We present $\lambda$-equitune and its special case equizero that outperform equitune on equivariant zero-shot and finetuning tasks. We show that both methods are equivariant, universal,

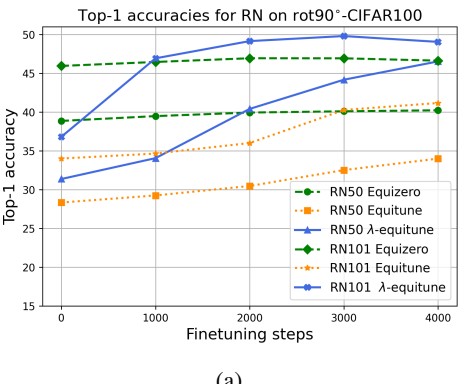
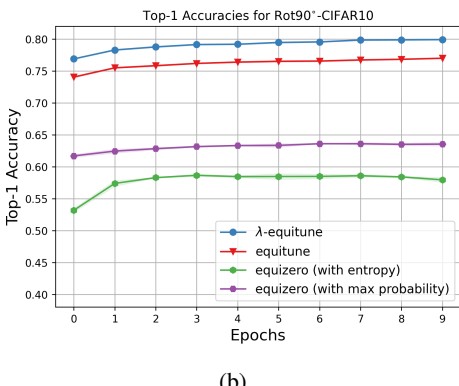

(a)                                              (b)

Figure 5: In (a) and (b), we plot accuracies of pretrained CLIP and Resnet, respectively, for equitune, equizero, and $\lambda$-equitune. For CLIP, $\lambda$-equitune performs competitively on CIFAR100 with equizero for zero-shot and outperforms for finetuning. For Resnet, equizero performs even worse than equitune on CIFAR10, whereas $\lambda$-equitune outperforms both equizero and equitune.

are computationally efficient, and are easy to implement. Equizero performs well when good proxy loss functions are available for downstream tasks, which we show is easy to find across several diverse tasks. $\lambda$-equitune outperforms equitune and is competitive with equizero without the need for any proxy loss. We consider diverse tasks: i) deep Q-learning, ii) fairness in NLG, iii) compositional generalization in languages, iv) CLIP-based classification, and v) CNN-based classification. Experimental results validate the superiority of $\lambda$-equitune and equizero over equitune on zero-shot and finetuning tasks.

## Acknowledgment

Discussion with Moulik Choraria on the $\lambda$-equitune finetuning for CLIP is appreciated. A portion of the work was supported by the Department of Energy (DOE) award (DE-SC0012704). This work was also supported in part by ZJU-UIUC Joint Research Center Project No. DREMES 202003, funded by Zhejiang University.

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
