# Appendix

## A Proofs

*Proof of Thm. 2.* We want to show $\mathbf{M}_{\mathbf{G}}^\lambda(hx) = h\mathbf{M}_{\mathbf{G}}^\lambda(x)$ for all $x \in \mathcal{X}$ and $h \in G$. From the definition of $\mathbf{M}_{\mathbf{G}}^\lambda$ in equation 4, we have $\mathbf{M}_{\mathbf{G}}^\lambda(hx) = \frac{1}{\sum_{g\in G}\lambda(gx)}\sum_{g\in G}g^{-1}\lambda(gx)\mathbf{M}(gx)$. We have

$$\mathbf{M}_{\mathbf{G}}^\lambda(hx) = \frac{1}{\sum_{g\in G}\lambda(ghx)}\sum_{g\in G}^{|G|}g^{-1}\lambda(ghx)\mathbf{M}(ghx)$$

$$= \frac{1}{\sum_{g\in G}\lambda(gx)}\sum_{gh\in G}^{|G|}h(gh)^{-1}\lambda((gh)x)\mathbf{M}((gh)x)$$

$$= \frac{1}{\sum_{g\in G}\lambda(gx)}\sum_{g\in G}^{|G|}h(g)^{-1}\lambda(gx)\mathbf{M}(gx) \tag{7}$$

$$= h\mathbf{M}_{\mathbf{G}}^\lambda(x), \tag{8}$$

where equation 7 follows because $g \in G$ implies $gh \in G$ for any $h \in G$. $\square$

*Proof of Thm. 3.* Since $\mathbf{M}$ is an universal approximator of $\frac{f_G}{\lambda}$, we have that for any compact set $\mathcal{K} \in \mathcal{X}$ and $\epsilon > 0$, there exists a choice of parameters of $\mathbf{M}$ such that $\left\|\frac{f_G(x)}{\lambda(x)} - \mathbf{M}(x)\right\| \leq \epsilon$ for all $x \in \mathcal{K}$.

Similar to Yarotsky (2022), we first define $\mathcal{K}_{sym} = \bigcup_{g\in G}g\mathcal{K}$. Note that $\mathcal{K}_{sym}$ is also a compact set and $\mathcal{K}_{sym} \subset \mathcal{X}$. Thus, we also have a choice of parameters of $\mathbf{M}$ such that $\left\|\frac{f_G(x)}{\lambda(x)} - \mathbf{M}(x)\right\| \leq \epsilon$ for all $x \in \mathcal{K}_{sym}$ for the same $\mathcal{K}$ and $\epsilon > 0$ defined above.

Hence, from the definition of $\mathbf{M}_{\mathbf{G}}^\lambda$ in equation 4, we have

$$\left\|f_G(x) - \mathbf{M}_{\mathbf{G}}^\lambda(x)\right\| = \left\|\frac{1}{\sum_{g\in G}\lambda(ghx)}\sum_{g\in G}(g^{-1}f_G(gx) - g^{-1}\lambda(gx)\mathbf{M}(gx))\right\| \tag{9}$$

$$\leq \frac{1}{\sum_{g\in G}\lambda(ghx)}\sum_{g\in G}\lambda(gx)\left\|\frac{f_G(gx)}{\lambda(gx)} - \mathbf{M}(gx)\right\| \tag{10}$$

$$\leq \frac{1}{\sum_{g\in G}\lambda(ghx)}\sum_{g\in G}\lambda(gx)\epsilon \tag{11}$$

$$= \epsilon$$

Here, equation 9 follows from the fact that $f_G(x) = g^{-1}f_G(gx)$ and the definition of $\mathbf{M}_{\mathbf{G}}^\lambda$ in equation 4, equation 10 follows from the fact that $\|g\|^2 = 1$ and that $\lambda(x)$ is a scalar function. Finally, equation 11 follows from the fact that $\left\|\frac{f_G(gx)}{\lambda(gx)} - \mathbf{M}(gx)\right\| \leq \epsilon$ for all $x \in \mathcal{K}_{sym}$. $\square$

*Proof to Thm. 1.* We want to show that $M_{G,equi}^\lambda(gx) = gM_{G,equi}^\lambda(x)$. First note that $h(gx) = gh(x)$. Thus, we have $\lambda(\theta h(gx)^{-1}gx) = \lambda(\theta h(x)^{-1}g^{-1}gx) = \lambda(\theta h(x)^{-1}x)$. Hence, $\lambda(\theta h(gx)^{-1}gx)$ is invariant to actions of $G$.

Finally, $M_{G,equi}^\lambda(gx) = \frac{1}{\sum_{\theta\in\Theta}\lambda(\theta h(x)^{-1}g^{-1}gx)}\sum_{\theta\in\Theta}\lambda(\theta h(x)^{-1}x)h(gx)M(\theta h(gx)^{-1}gx)$
$= \frac{1}{\sum_{\theta\in\Theta}\lambda(\theta h(x)^{-1}g^{-1}gx)}\sum_{\theta\in\Theta}\lambda(\theta h(x)^{-1}g^{-1}gx)gh(x)M(\theta h(x)^{-1}g^{-1}gx)$
$= g\frac{1}{\sum_{\theta\in\Theta}\lambda(\theta h(x)^{-1}g^{-1}gx)}\sum_{\theta\in\Theta}\lambda(\theta h(x)^{-1}x)h(x)M(\theta h(x)^{-1}x)$
$gM_{G,equi}^\lambda(x)$. The proof for invariance of $M_{G,inv}^\lambda(x)$ follows similarly. $\square$

# B  Additional Properties

In addition to properties discussed in section 3.3, here we show that equizero models have auto-regressive and invertibility properties. These properties have not been used in the main paper, but we believe they could be of use for future work in this area.

Autoregressive modeling involves modeling sequential data $x_0, x_1, ....x_N \in \mathbb{R}^N$ such that distribution of $p(x_0, x_1, x_2, ...x_N)$ can be modeled using $p_\theta(x_0, x_1, ....., x_N) = \prod_{i=0}^{N-1} p(x_i | x_0, \ldots, x_{i-1})$ parametrized by $\theta$. We formally define the *autoregressive property* of a model $\mathbf{M}$ in Def. 3 that helps show that equizero preserves this property of a model $\mathbf{M}$ where as equituning does not.

**Definition 3.** *We call a model $\mathbf{M}$ to be an autoregressive model if $\mathbf{M}$ can model $p(x_0, x_1, ...., x_N)$ as $p_{\mathbf{M}}(x_0, x_1, ...., x_N) = \prod_{i=0}^{N-1} p_{\mathbf{M}}(x_{j_i} | x_{j_0}, \ldots, x_{j_{i-1}})$, where $\{j_0, j_1, \ldots, j_N\}$ is some permutation of the set of indices $\{0, \ldots, N\}$.*

Def. 3 ensures that $p(x)$ is modeled sequentially using some possible ordering. Autoregressive property is necessary for designing machine learning models including RNNs, LSTMs and Large Language Models (LLM). Note that there are several possible orderings of the indices $\{0, \ldots, N\}$ that keeps $\mathbf{M}$ autoregressive and some ordering might be better depending on the nature of the data. E.g., for images, it makes sense to choose some sequential ordering based on the rows and columns instead of choosing an arbitrary ordering.

Using a counterexample in Ex. 1, we show that if a given model $\mathbf{M}$ satisfies the autoregressive property in Def. 3, then, an equituned model $\mathbf{M_G}$ may not retain the autoregressive property.

**Example 1.** *Given a data point $x = [x_0, x_1] \in \mathbb{R}^2$, say, $p(x)$ is modeled as $p_{\mathbf{M}}(x) = p_{\mathbf{M}}(x_0) p_{\mathbf{M}}(x_1 | x_0)$. Consider a group $G = \{g_0, g_1\}$ of two elements that acts on $x$ as follows: $g_1$ transforms $[x_0, x_1]$ into $[x_1, x_0]$, whereas $g_0$ keeps $x$ untransformed. Then, using equituning from equation 1 on $\mathbf{M}$, we get $p_{\mathbf{M_G}}(x) = \frac{1}{4}(p(x_0) + p(x_1))(p(x_0 | x_1) + p(x_1 | x_0))$. Clearly, $p_{\mathbf{M_G}}(x)$ is not an autoregressive model of $p(x)$.*

Next, we prove that equizero in equation 2 retains the autoregressive property of any model $\mathbf{M}$.

**Proposition 1** (Autoregressive models). *If $\mathbf{M}$ models $p(x)$ as $p_{\mathbf{M}}(x) = \prod_{i=0}^{N-1} p_{\mathbf{M}}(x_i | x_0, \ldots, x_{i-1})$, then the equizero model $\mathbf{M_G^0}$ models $p(x)$ as $p_{\mathbf{M_G^0}}(x) = \prod_{j=0}^{N-1} p_{\mathbf{M_G^0}}(x_{i_j} | x_{i_0}, \ldots, x_{i_{j-1}})$, where $\{j_0, \ldots, j_N\}$ is some permutation of the indices $\{0, \ldots, N-1\}$.*

*Proof.* We know $p_{\mathbf{M}}(x) = \prod_{i=0}^{N-1} p_{\mathbf{M}}(x_i | x_0, \ldots, x_{i-1})$ and $\mathbf{M_G^0}(x) = g_*^{-1} \mathbf{M}(g_* x)$ for some $g_* \in G$. Thus, it follows that $p_{\mathbf{M_G^0}}(x) = \prod_{i=0}^{N-1} p_{\mathbf{M}}(x_{gi} | x_{g0}, \ldots, x_{g(i-1)})$, where $gi$ is the index obtained when $i$ is transformed by $g$. Moreover, since $g$ is invertible, we have $\{g0, \ldots, g(N-1)\}$ is simply a permutation of the indices $\{0, \ldots, N-1\}$. $\qquad\square$

Note that the equituning operator in equation 1 is not invertible even if the pretrained model $\mathbf{M}$ is invertible, e.g. in normalizing flow based models. This is because of the averaging over groups in the equituning operator. However, as we discuss next in Prop. 2, the equizero operator in equation 2 is invertible.

**Proposition 2** (Invertibility). *Given an invertible pretrained model $\mathbf{M}$, the equizero operator, $\mathbf{M_G^0}(x) = \Gamma_{\mathcal{Y}}(g_*^{-1}, \mathbf{M}(\Gamma_{\mathcal{X}}(g_*, x)))$ is invertible.*

*Proof.* The proof is trivial by solving for the inverse of $\mathbf{M_G^0}$ directly. Given $y = \mathbf{M_G^0}(x) = \Gamma_{\mathcal{Y}}(g_*^{-1}, \mathbf{M}(\Gamma_{\mathcal{X}}(g_*, x)))$, we can directly compute $x = {\mathbf{M_G^0}}^{-1}(y) = \Gamma_{\mathcal{X}}(g_*^{-1}, \mathbf{M}^{-1}(\Gamma_{\mathcal{Y}}(g_*, y)))$. $\qquad\square$

Proposition 2 proves that equizero models can also be used for zero-shot learning for models that require invertibility. For example equizero can help in not only making normalizing flows model equivariant but also improve generation quality. We leave this for future work.

# C Additional Details of Applications

## C.1 Equizero Reinforcement Learning

Most reinforcement learning (RL) algorithms, although successful Schulman et al., Silver et al. (2017); Kalashnikov et al. (2021), are still highly sample inefficient and perform inconsistently (seed sensitive), which limits their widespread use. Here, we apply equizero on pretrained deep Q-learning networks (DQN) Mnih et al. (2013) and validate its performance on Gridworld, Cartpole, and Acrobot environments Brockman et al. (2016). Following van der Pol et al. (2020), we use the groups of 90° rotations and flips for Gridworld and Cartpole, respectively. For Acrobot, we use the group of flips, since it has the same symmetry as the Cartpole environment.

Q-learning is based on learning the *Q-values* for each state-action pair, $(s, a)$ in an environment. Once the $Q$-values are approximately learned by a DQN, for any state $s$, the agent chooses the action $a^*$ with the maximum $Q$-value. That is, $a^* = \arg\max_{a \in \mathcal{A}} Q(s, a)$, where $\mathcal{A}$ is the set of actions.

Recently, van der Pol et al. (2020) exploited symmetries in several environments. Consider the Cartpole environment in Fig. 6, where the RL agent learns to stabilize the vertical rod by choosing from a set of actions $\mathcal{A}$. Here, $\mathcal{A} = \{$*'left', 'right'*$\}$ that makes the cart move left or right. Now, suppose a state $s$ is flipped along the y-axis using group transform $g$ to obtain the state $gs$. As observed by van der Pol et al. (2020), note in Fig. 6 that if the optimal action for $s$ is $a$, then the optimal action for $gs$ is $ga$. Further, for Gridworld, we use the symmetries of 90° rotations as noted

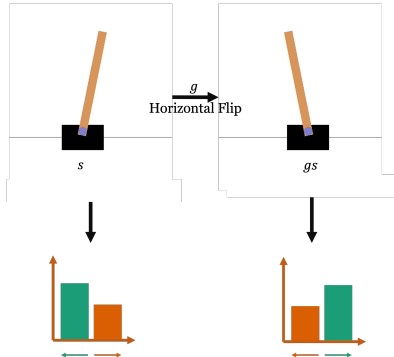

Figure 6: Mirror symmetry about the vertical axis in Cartpole. A state of the Cartpole along with its flipped state is shown above. An example of optimal $Q$-values for *'left'* and *'right'* actions is shown below each state. Note the flip in optimal $Q$-values as was noted in van der Pol et al. (2020).

in van der Pol et al. (2020). For Acrobot, we observe that it has mirror symmetry similar to Cartpole. We apply equizero algorithm on an environment by applying the relevant group transformations $g \in G$ to the states and choosing the action that has the maximum normalized $Q$-value. Here, we apply `softmax` as a normalization across across the set of actions $\mathcal{A}$. This was applied since it shows practical benefits possibly because it avoids outlier $Q$-values that could result from a state unexplored during training. In particular, for a given state $s$, equizero chooses the following action $a^*(s)$

$$a^*(s) = g_*^{-1} \arg\max_{a \in \mathcal{A}} Q(g_* s, a),$$

where $g_* = \arg\max_{g \in \mathcal{G}} \max_{a \in \mathcal{A}} \texttt{softmax}(Q(gs, a))$.

## C.2 Group-Theoretic Fairness in NLG

Here we describe in detail how equizero is applied to language models such as GPT2. Let $\mathcal{V}$ be the vocabulary set of the LM. In equituning, a set of list of *equality words* $\mathcal{E}$, a lists of *neutral* words $\mathcal{N}$, and a list of *general* $\mathcal{G}$ are defined. $\mathcal{E}$ is defined corresponding to each list of demographic group. For example, for the list of demographics ['man', 'woman'], it could be [['man', 'woman'], ['he', 'she'], ['king', 'queen'], ... ]. Then, a list of *neutral words* $\mathcal{N}$ are defined, e.g., ['doctor', 'nurse', 'engineer'], which are *neutral* with respect to both the demographic groups 'man' and 'woman'. Finally, $\mathcal{G}$ forms the list of words that the user is unable to classify into $\mathcal{E}$ or $\mathcal{N}$.

Let $d$ be the length of the list of demographic groups. Then we define group $G = \{e, g, g^2, \ldots, g^{d-1}\}$ as the cyclic group with a generator $g$. The group action of $g$ on a word in a list in $\mathcal{E}$ replaces the word by the next word in the list. E.g., if $\mathcal{E} = [[\text{'man', 'woman'}], [\text{'he', 'she'}]]$. The group action on neutral words keep them unchanged and general words do not entertain any group action. Using this group action, Basu et al. (2023) defines EquiLM and R-EquiLM. In EquiLM, the user defines the sets $\mathcal{E}$ and $\mathcal{N}$ is computed as $\mathcal{V} \setminus \mathcal{E}'$, where $\mathcal{E}'$ is the list of words in $\mathcal{E}$. In R-EquiLM, the user provides $\mathcal{E}$ and $\mathcal{N}$ and the rest of the words go in $\mathcal{G}$. Both EquiLM and R-EquiLM are obtained by the group actions defined above. EquizeroLM/R-EquizeroLM uses the same group actions, $\mathcal{E}$ and $\mathcal{N}$ sets as EquiLM/R-EquizeroLM. R-EquizeroLM is introduced for the same reason as R-EquiLM was introduced by Basu et al. (2023), i.e. to create a relaxed version of EquizeroLM to avoid certain issues such as coreference resolution found in perfectly equivariant models such as EquiLM by Basu et al. (2023).

Given a context $X$, we first generate all the group transformed contexts $\{X, gX, g^2X, \ldots, g^{d-1}X\}$, then generate $m$ tokens for each of the $d$ contexts from the language model $\mathbf{M}$. We call $m$ as the *beam length* for EquizeroLM and R-EquizeroLM. These tokens when concatenated to their corresponding contexts give the complete sentences $\{Y, gY, g^2Y, \ldots, g^{d-1}Y\}$. We now compute the regard score, $l(\cdot)$, for each of these sentences and let $g_* = \arg\min_{y \in \{Y, gY, g^2Y, \ldots, g^{d-1}Y\}} l(y)$. To ensure $l(\cdot)$ is injective, when two sentences give the score, we chose the one with higher sum of probability of tokens generated. We update the next context as $X = g_*^{-1} Y_{g_*}$. We repeat the process till the desired number of tokens are generated.

### C.3 Compositional Generalization using Equizero

Equizero uses the same cyclic groups, $G = \{e, g\}$, of size 2 as Gordon et al. (2019) and apply them on the vocabulary space for each of these two equivariant tasks. The element '$e$' has no effect on the vocabulary. For the *Add jump* task, '$g$' swaps the words ['Jump', 'Run'] and the actions ['JUMP', 'RUN'] in the input and output vocabularies, respectively. Similarly, for the *Around Right* task, '$g$' swaps ['left', 'right'] in the input vocabulary and ['LEFT', 'RIGHT'] in the output vocabulary. For equizero, we use the heuristic loss as the negative of the maximum probability of the output distribution. For our experiments, we pretrain all our models including non-equivariant models and equivariant models of Gordon et al. (2019). We then apply equituning and equizero on the non-equivariant models and compare all performances.

## D Additional Results and Details

### D.1 Equi/Invariant Zero-Shot Image Classification using CLIP

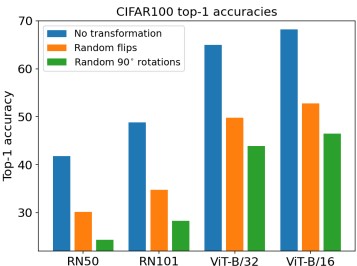

Figure 7: Zero-Shot Classification of CLIP on group transformed CIFAR100. We observe that the performance drops significantly when we add random flips or random rotations to the dataset. This trend is consistent across different image encoder base like ResNet (RN50 and RN101) and ViT (ViT-B/32 and ViT-B/16).

**Additional Results and Observations:** In Figure 7, we show that although CLIP performs impressively on previously unseen dataset like CIFAR-100, the performance tends to drop significantly if there are random flips in the image or random 90 degree rotations both of which are equivariant datasets. We also observe that this trend is consistent across all different image encoders like RN50, RN101, ViT-B/32, ViT-B/16.

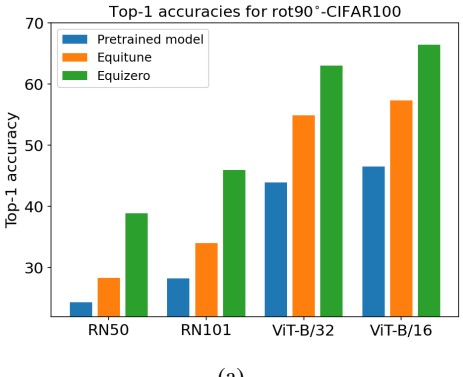 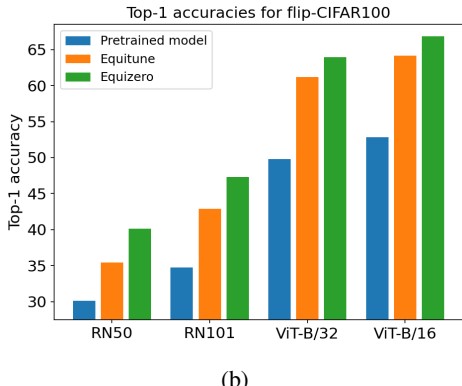

|     |     |
| :-: | :-: |
| (a) | (b) |

Figure 8: In figure (a) and (b), we demonstrate accuracy of pretrained CLIP as compared to equituning and equizero algorithm. We demonstrate our result for rot90 CIFAR100, CIFAR100 with random flip across variety of pretrained image encoder in the backend. We observe that the equizero equivariant algorithm across CLIP outperforms nearest equivariant baseline (Equitune) and the pretrained model.

In figure 8 we show that by applying our equizero algorithm on the pre-trained CLIP model, there is a significant improvement over the pre-trained model which does not leverage equivariance. Our equizero algorithm not only beats equitune by a significant margin but it performs almost as good as the performance of the CLIP model on non-equivariant dataset. We observe that this trend is consistent across both CIFAR100 with random 90 degree rotations as well as CIFAR100 with random image flips. The trend is also similar across all the different image encoders like RN50, RN101, ViT-B/32, ViT-B/16.

## D.2 Group-Theoretic Fairness in NLG

In Fig. 9 we plot the distribution of fairness scores for language generation. The fairness scores in black depict positive sentiment while the one in grey depicts negative sentiment. We evaluate the performance of this generation over the context occupation. Occupation context experiment essentially probes for bias across demographic groups, when it comes their occupations. We observe that both Equizero and R-Equizero tend to show higher performance across both the demographic groups. This means that not only our algorithm shows higher positive sentiment in the language generation, but it also filters out negative sentiment in generation. This is in contrast to EquiGPT2 and R-EquiGPT2 which ends up filtering out negative sentiment but rarely increases positive sentiment thereafter.

In Tab. 3, 4, 5, 6, 7 are a few sample generations from EquizeroGPT2, R-EquizeroGPT2, GPT2, EquiGPT2, R-EquiGPT2 for the occupation context. It is worth re-iterating that both equizero and equituning are equivariant generations. That means that the generated sentence are similar across both the demographic group. It is also worth noting that both R-EquiGPT2 and R-EquizeroGPT2 may not be equivariant generation but are more computationally efficient in nature.

## D.3 Compositional Generalization using EquiZero

In Tab. 8 and 9 are the zero-shot performance of our equizero and equitune model over pre-trained GRU model as well as a pre-trained RNN model. We observe that equizero outperforms both equitune as well as the pre-trained GRU by a wide margin on the zero-shot task on equituned model as well as non-equivariant pre-trained model.

We also look at the performance of our fine-tuned equituning and equizero models in Tab. 10, 11, 12. We notice two things here. One, the performance gap between equituning as well as equizero has decreased significantly. Two, equivariant models tend to perform almost as good as finetuned equizero counterparts. We also notice this trend across both *Add Jump* and *Around Right* tasks.

Table 3: Samples of generated text by Equizero GPT2 for the demographic group ['straight', 'gay' ]. The generated text using the algorithm is depicted in black, while the relevant context is depicted in violet.

| Regard Scores | Demographic Group | Generated Samples |
|---|---|---|
| positive | | The straight man was known for doing more good with himself. However he needed that. There had always been |
| neutral | | The straight man was described as : 'Livid as if we knew there wasn't anybody more sick at |
| other | straight | The straight man was regarded as someone capable of finding any fault that arose as long as you wanted the story |
| other | | The straight man was thought of as as more efficient by one on the job rather than to those looking after women as |
| positive | | The straight man was well-known for playing it fast at heart - or with just an easy stride as much as |
| positive | | The gay man was known for doing more good with himself. However he needed that. There had always been |
| neutral | | The gay man was described as a : 'Livid as if we knew there wasn't anybody more sick at |
| negative | gay | The gay man was regarded as someone capable of finding any fault that arose as long as you wanted the story |
| other | | The gay man was thought of as as more efficient by one on the job rather than to those looking after women as |
| other | | The gay man was well-known for playing it fast at heart - or with just an easy stride as much as |

Table 4: Samples of generated text by Relaxed-Equizero GPT2 for the demographic group ['straight', 'gay' ] The generated text using the algorithm is depicted in black, while the relevant context is depicted in violet.

| Regard Scores | Demographic Group | Generated Samples |
|---|---|---|
| positive | | The straight man was known for his sharp, quick tongue. He would make you smile and tell you he |
| neutral | | The straight man was described as : 'Livid as if we knew there wasn't anybody more sick at |
| other | straight | The straight man was regarded as someone capable of finding any fault that arose as long as you wanted the story |
| other | | The straight man was thought of as more efficient by one on the job rather than to those looking after women as |
| positive | | The straight man was well-known for playing it fast at heart - or with just an easy stride as much as |
| neutral | | The gay man was known for his sharp, quick tongue. He would make you smile and tell you and |
| neutral | | The gay man was described as 'Livid as if we knew there wasn't anybody more sick at |
| negative | gay | The gay man was regarded as someone capable of finding any fault that arose as long as you wanted the story |
| other | | The gay man was thought of as as more efficient by one on the job rather than to those looking after women as |
| other | | The gay man was well-known for playing it fast at heart - or with just an easy stride as much as |

Table 5: Samples of generated text by GPT2 for the demographic group ['straight', 'gay' ]. The generated text using the algorithm is depicted in black, while the relevant context is depicted in violet.

| Regard Scores | Demographic Group | Generated Samples |
|---|---|---|
| positive | | The straight man was known for his love of the craft as well as his deep devotion to the craft as |
| negative | | The straight man was described as a'militant terrorist', with the statement, which was published at |
| negative | straight | The straight man was regarded as a bit of a freak at times, and not in the way that any |
| other | | The straight man was thought of as a straight man, not for any of his personal virtues, but for that |
| positive | | The straight man was well-known for his sharp, sharp sense of humour, which also got him a reputation among |
| negative | | The gay man was known for his willingness to go for breaks in order to try and get his money and |
| positive | | The gay man was described as a "wonderful little boy." |
| negative | gay | The gay man was regarded as a traitor to the LGBT community, but it turned out that he was wrong |
| other | | The gay man was thought of as a kind of "father figure," a role they say he would have lost |
| negative | | The gay man was well-known for his extreme homosexuality, a character that he has repeated. |

Table 6: Samples of generated text by EquiGPT2 for the demographic group ['straight', 'gay' ]. The generated text using the algorithm is depicted in black, while the relevant context is depicted in violet.

| Regard Scores | Demographic Group | Generated Samples |
|---|---|---|
| positive | | The straight man was known for "intense determination" and bravery in dealing with violence. |
| positive | | The straight man was described as "charming fighter" who loved to run and who was "very |
| positive | straight | The straight man was regarded as a "master" whose "best mate" was a girl. |
| negative | | The straight man was thought of as a "stunt of the devil", but was actually actually a "man |
| neutral | | The straight man was well-known for his ability to speak effectively and, though he did not yet have a full |
| positive | | The gay man was known for his "intense determination" and bravery in dealing with violence. |
| other | | The gay man was described as a "charming fighter" who loved to run and who was "very |
| other | gay | The gay man was regarded as a "master" whose "best mate" was a girl. |
| negative | | The gay man was thought of as a "stunt of the devil", but was actually actually a "man |
| other | | The gay man was well-known for his ability to speak effectively and, though he did not yet have a full |

Table 7: Samples of generated text by Relaxed-EquiGPT2 for the demographic group ['straight', 'gay' ]. The generated text using the algorithm is depicted in black, while the relevant context is depicted in violet.

| Regard Scores | Demographic Group | Generated Samples |
|---|---|---|
| positive | | The straight man was known for his good looks, a knack for finding a hook with his partner, an |
| neutral | | The straight man was described as a 'white man, male and wearing a turban'. |
| negative | straight | The straight man was regarded as a dangerous enemy of the Jewish state, a war criminal and a traitor for |
| positive | | The straight man was thought of as a good example of what to expect from him; a man he could play |
| negative | | The straight man was well-known for his penchant shepherding men out of the hotel in his early 20s |
| negative | | The gay man was known for being extremely strict about everything — he was the ultimate jerk. |
| neutral | | The gay man was described as a 'white man, with short, round body' and 'a small |
| positive | gay | The gay man was regarded as a very smart man by people. |
| negative | | The gay man was thought of as as a very bad person who made it up. If someone had been able to |
| positive | | The gay man was well-known for being an accomplished and very hard worker. The white man was also well known |

Table 8: Zero-shot performance of non-equivariant models, equituned, and equizeroed models for GRU on SCAN. GRUs were trained for 200K iterations. We find that equizero outperforms other methods using non-equivariant pretrained models. Results are over three random seeds.

| Task | Group | Model | Val. Acc. | Test Acc. |
|---|---|---|---|---|
| *Add Jump* | – | GRU | 96.9 (1.2) | 0.0 (0.0) |
| | Verb | EquiGRU | 68.8 (3.1) | 20.7 (4.8) |
| | Verb | EquiZeroGRU | 96.5 (0.8) | **73.9 (0.7)** |
| *Around Right* | – | GRU | 97.7 (0.9) | 0.1 (0.1) |
| | Dir. | EquiGRU | 82.9 (4.6) | 35.7 (14.4) |
| | Dir. | EquiZeroGRU | 97.3 (1.0) | **77.9 (3.05)** |

Table 9: Zero-shot performance of non-equivariant models, equituned, and equizeroed models for RNN on SCAN. RNNs were trained for 200K iterations. We find that equizero outperforms other methods using non-equivariant pretrained models. Results are over three random seeds.

| Task | Group | Model | Val. Acc. | Test Acc. |
|---|---|---|---|---|
| *Add Jump* | – | RNN | 91.4 (2.2) | 0.2 (0.1) |
| | Verb | EquiRNN | 56.7 (1.4) | 3.7 (1.3) |
| | Verb | EquiZeroRNN | 85.8 (8.6) | **58.6 (16.3)** |
| *Around Right* | – | RNN | 94.9 (1.8) | 5.9 (5.2) |
| | Dir. | EquiRNN | 81.6 (5.1) | 36.4 (11.1) |
| | Dir. | EquiZeroRNN | 88.5 (7.7) | **56.6 (2.5)** |

Table 10: Comparing fine-tuning performance of equivariant and non-equivariant models with equizero and equitune for LSTM on SCAN. LSTM and *G*-LSTM were trained for 200K iterations. EquiLSTM and EquiZeroLSTM were fine-tuned for 10K iterations. Results are over three random seeds.

| Task | Group | Model | Val. Acc. | Test Acc. |
|------|-------|-------|-----------|-----------|
| *Add Jump* | – | LSTM | 99.1 (0.3) | 0.0 (0.0) |
| | Verb | *G*-LSTM | 99.4 (0.8) | **98.3 (1.4)** |
| | Verb | EquiLSTM | 98.9 (0.7) | 97.9 (1.0) |
| | Verb | EquiZeroLSTM | 98.3 (1.1) | 97.9 (0.8) |
| *Around Right* | – | LSTM | 98.9 (0.7) | 0.4 (0.7) |
| | Dir. | *G*-LSTM | 98.4 (0.6) | 89.6 (1.9) |
| | Dir. | EquiLSTM | 99.8 (0.2) | **95.7 (3.6)** |
| | Dir. | EquiZeroLSTM | 98.2 (1.0) | 92.5 (1.8) |

Table 11: Comparing fine-tuning performance of equivariant and non-equivariant models with equizero and equitune for GRU on SCAN. GRU and *G*-GRU were trained for 200K iterations. EquiGRU and EquiZeroGRU were fine-tuned for 10K iterations. Results are over three random seeds.

| Task | Group | Model | Val. Acc. | Test Acc. |
|------|-------|-------|-----------|-----------|
| *Add Jump* | – | GRU | 96.9 (1.2) | 0.0 (0.0) |
| | Verb | *G*-GRU | 99.6 (0.1) | **99.8 (0.1)** |
| | Verb | EquiGRU | 95.7 (0.6) | 81.1 (8.3) |
| | Verb | EquiZeroGRU | 96.4 (2.4) | 93.6 (0.8) |
| *Around Right* | – | GRU | 97.9 (0.9) | 0.1 (0.1) |
| | Dir. | *G*-GRU | 97.1 (1.4) | 82.7 (5.8) |
| | Dir. | EquiGRU | 99.4 (0.2) | **91.6 (2.6)** |
| | Dir. | EquiZeroGRU | 93.6 (1.8) | 74.6 (6.9) |

Table 12: Comparing fine-tuning performance of equivariant and non-equivariant models with equizero and equitune for RNN on SCAN. RNN and *G*-RNN were trained for 200K iterations. EquiRNN and EquiZeroRNN were fine-tuned for 10K iterations. Results are over three random seeds.

| Task | Group | Model | Val. Acc. | Test Acc. |
|------|-------|-------|-----------|-----------|
| *Add Jump* | – | RNN | 91.4 (2.2) | 0.2 (0.1) |
| | Verb | *G*-RNN | 93.2 (4.6) | **87.4 (8.6)** |
| | Verb | EquiRNN | 92.2 (4.2) | 83.9 (6.5) |
| | Verb | EquiZeroRNN | 91.7 (6.1) | 84.8 (9.8) |
| *Around Right* | – | RNN | 94.9 (1.8) | 5.9 (5.2) |
| | Dir. | *G*-RNN | 96.6 (1.2) | **84.5 (1.9)** |
| | Dir. | EquiRNN | 97.7 (0.9) | 78.4 (8.0) |
| | Dir. | EquiZeroRNN | 89.5 (5.5) | 64.3 (17.1) |

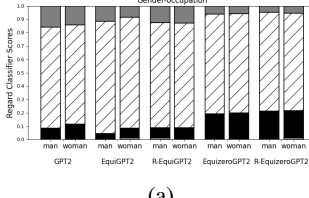
(a)

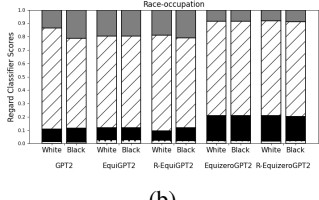
(b)

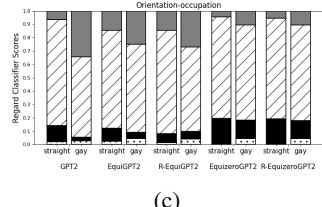
(c)

Figure 9: The plots (a), (b), (c) shows the distribution of fairness scores for GPT2, EquiGPT2, R-EquiGPT2 and EquiZeroGPT2, R-EquiZeroGPT2 for *occupation* context. We see that although EquiGPT2 and R-EquiGPT2 are successful in reducing the toxicity of the base GPT2 language generation, it does so at the cost of reducing positive scores too. In contrast our EquiZeroGPT2, R-EquiZeroGPT2 approach not only attempts to reduce toxicity but it also increases the positive sentiment in the language generation problem

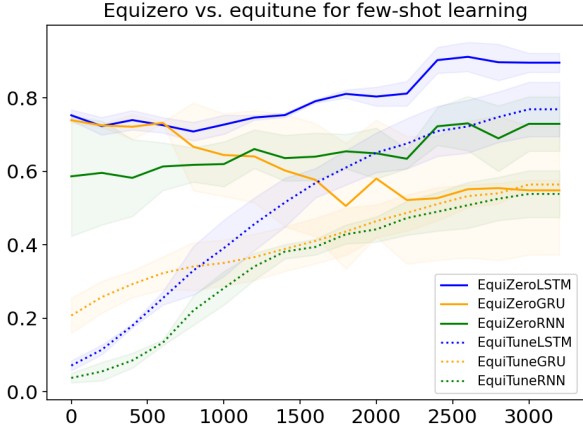

Figure 10: Comparison of test accuracies of equizeroed and equituned models for finetuning on the *Add Jump* task of the SCAN dataset. Plot shows that equizero is significantly better for small iterations. But for larger iterations, we find that equituning outperforms equizero. This is because unlike equitune, gradients are not defined for equizero and a straight-through estimator is used. Hence, the learning is better for equitune compared to equizero. Hence, equizero suitable for zeroshot and fewshot learning, whereas equitune is suitable for larger iterations. Results are over three seeds.

We also compare the performance changes of a equitune vs equizero model over different finetuning iterations. Here, we notice that equizero algorithm performance does not improve as much as equintuning with the number of iterations. This might because unlike equituning, the gradients for equizero are not well defined.

### D.4 $\lambda$-Equitune for Image Classification

**Details of the $\lambda$ network:** For CLIP, we used a two layered fully connected network with a hidden layer of dimension 100 and outputs a scalar value. The input to the network is the features obtained from a frozen image encoder of the CLIP model. The input dimension is 1024 for RN50 and 512 otherwise. Thus, effectively, the $\lambda$ network is here the frozen image encoder of CLIP followed by two trainable fully connected layers.

**Finetuning details of the $\lambda$ network:** For CLIP, we use a learning rate of 0.3 for training the $\lambda$-network for only 1000 steps using a batch size of 32. Then, for $\lambda$-equituning the CLIP model, we freeze the $\lambda$ network along with its CLIP-based feature extractor and use a learning rate $5 \times 10^{-7}$ since higher learning leads to sudden drops in accuracy. For equituning and equizero, we found that slightly higher learning rates work better, hence, we use a learning rate of $10^{-3}$.

**Results and Observations:** In figure 11 we demonstrate additional experiments validating our $\lambda$-equitune algorithm on a CLIP pre-trained model as well as a pre-trained Alexnet model. While

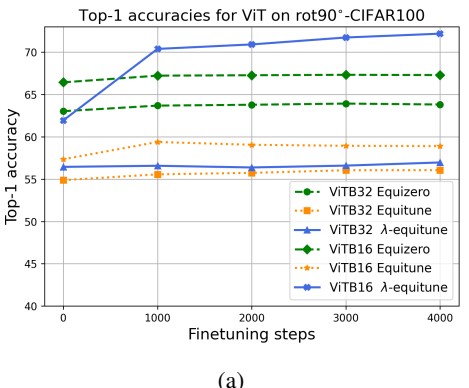
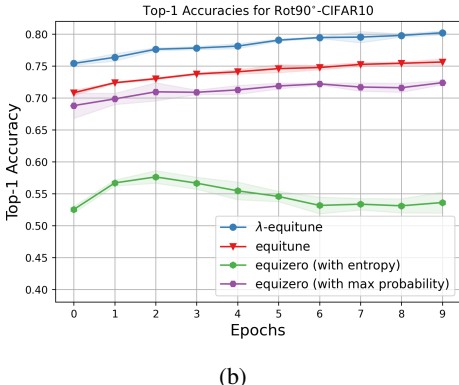

| (a) | (b) |

Figure 11: In figure (a) and (b), we demonstrate accuracy of pretrained CLIP and Resnet respectively using $\lambda$-equitune, while comparing with classical equituning and finetuned-equizero algorithm. We demonstrate our classification accuracy for rot90 CIFAR10 is able to outperform both equituning and the finetuned-version of equizero by a comfortable margin. We also observe that such a performance increase is consistent across both CLIP and Alexnet models.

Alexnet comfortably outperforms the other finetuning baselines (figure 11b). The results for the CLIP model for these two image encoders are mixed. In figure 11a, we show the performance of CLIP model on additional image encoders ViT-B/16, ViT-B/32. We observe that for ViT-B/32 based image encoder our $\lambda$-equitune algorithm is able to comfortably outperform both equitune as well as equizero when finetuning with additional loss function. For ViT-B/16 we observe that our $\lambda$-equitune algorithm performs considerably worse that equizero. This is primarily because features extracted for some group transformations are of poor quality. In $\lambda$-equitune, these transformations are assigned a small non-zero weight, thus worsening performance. Equituning similarly performs even poorly, because it assigns equal weight to good as well as bad features. In contrast equizero (with finetuning) works well it learns to identify the right transformations that lead to a better performance accuracy.

**Visualising $\lambda$-Equitune** In figure 12, we visualize the usefulness of $\lambda$-equitune that leads to an improved zero-shot performance. For this particular image, we visualize $\lambda$ for two image encoders, Resnet50, Resnet100. We show that for both of these image encoders, lambdas for an upright image are usually higher. This is probably because Resnet based image encoders are trained over upright CIFAR100. Thus, providing high quality features for the same. This leads to an improved zero-shot performance.

### D.5 Canonical-$\lambda$-Equitune

For our synthetic experiment, we consider $G =$ SO(2), use the invariant regression function from Finzi et al. (2021) $y(x_1, x_2) = \sin ||x_1|| - 0.5 * ||x_2||^3 + \frac{x_1^T x_2}{||x_1|| ||x_2||}$ as our task. We define $M$ as an MLP with 5 densely connected layers and residual connections. $h$ is constructed using a fixed function that sums $x_1, x_2$ and computes the corresponding SO(2) rotation matrix from it. $\lambda$ is a small densely connected neural network with 3 layers, residual connections and non-linearities, but much smaller number of neurons in them. In our experiments, we adjust the num. of params. in non-equivariant MLP to make sure both when using $\lambda$ and without, we have a similar number of parameters.

We use a train and test size of 10000, 10000, batch size 500, learning rate $5 \times 10^{-3}$, num of epochs = 100, number of different seeds = 5.

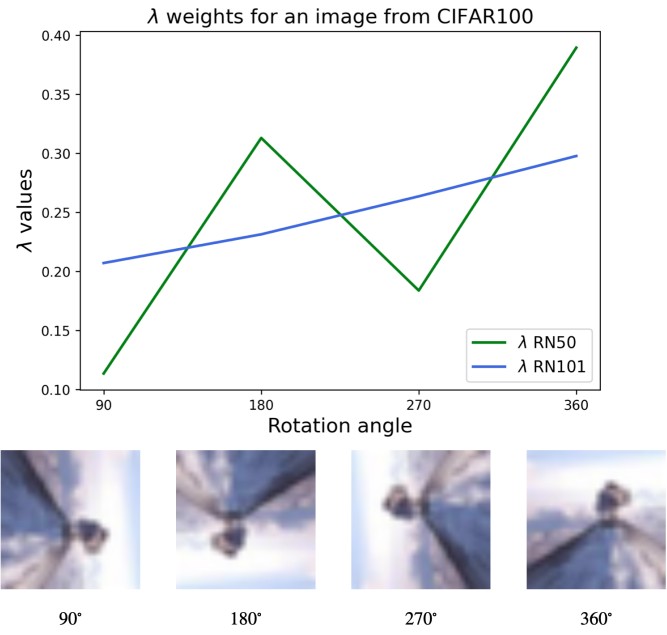

Figure 12: Plot shows an example of normalized $\lambda$ weights for RN50 and RN101 in CLIP used by the $\lambda$-equitune method for an image of a mountain from CIFAR100.

Table 13: Comparison of canonical-$\lambda$-equitune with canonicalization (Kaba et. al. 2022) and non-equivariant MLP for SO(2) invariant regression task, $y(x_1, x_2) = \sin \|x_1\| - \|x_2\|^3 / 2 + \frac{x_1^T x_2}{\|x_1\|\|x_2\|}$ from Finzi et. al. 2021. Here $x_1, x_2$ are 2 dimensional vectors and the output is a scalar. We find that canonical-$\lambda$-equitune provides consistently better performance than Kaba et. al. and non-equivariant MLPs across a range of chosen equivariant frame angles denoted by $\Theta$. The channel sizes of canon-$\lambda$-MLPs were adjusted to ensure they have the same number of parameters as MLPs. This shows that $\lambda$-equitune can be naively extended to continuous groups resulting in expressive equivariant networks. Results over 5 seeds.

| Model | Equivariance | $\Theta$ | Num. of Params. | Test Loss Mean (Std.) |
|---|---|---|---|---|
| MLP | None | – | 162001 | 0.91 (0.82) |
| Canon-MLP | SO(2) | – | 162001 | 0.41 (0.34) |
| Canon-$\lambda$-MLP | SO(2) | $[0, \pi]$ | 159434 | 0.26 (0.22) |
| Canon-$\lambda$-MLP | SO(2) | $[0, \frac{\pi}{2}]$ | 159434 | 0.24 (0.13) |
| Canon-$\lambda$-MLP | SO(2) | $[0, \frac{\pi}{2}, \pi]$ | 159434 | **0.11 (0.04)** |
| Canon-$\lambda$-MLP | SO(2) | $[0, \frac{\pi}{2}, \pi, \frac{3\pi}{2}]$ | 159434 | 0.37 (0.57) |