# OpenReview forum: "Efficient Equivariant Transfer Learning from Pretrained Models"
_NeurIPS.cc/2023/Conference — NeurIPS 2023 poster_

### Official Review · Reviewer_8HWc · 2023-07-05

**Soundness:** 3 good
**Presentation:** 3 good
**Contribution:** 3 good
**Rating:** 7
**Confidence:** 3

**Summary:**

The paper introduces $\lambda$-equitune, an innovative method that refines existing strategies for achieving equivariant outputs from non-equivariant neural networks. $\lambda$-equitune employs importance weights for feature averaging, which outperforms the group averaging method equine. The authors also present equizero, another approach enhancing zero-shot and fine-tuned performance. The effectiveness of these methods is validated across diverse applications and models, including image classification, deep Q-learning, and natural language generation.

**Strengths:**

**Originality**: The paper introduces a novel approach to addressing the limitations of existing equivariance methods by incorporating importance weights for feature averaging. This creative combination of ideas contributes to the originality of the work.

**Quality**: The research is of high quality, backed by rigorous theoretical justifications and empirical evaluations. The authors provide compelling evidence to support the claims made, ensuring the reliability and robustness of their proposed method.

**Clarity**: The paper is exceptionally well-written, presenting complex concepts in a clear and concise manner. The organization of the paper facilitates understanding, and the inclusion of theoretical proofs enhances its clarity.

**Significance**: The significance of the paper lies in its broad applicability across multiple domains and models, including image classification, deep Q-learning, and natural language generation. The improved zero-shot and fine-tuned results achieved by $\lambda$-equitune and equizero highlight their potential to advance the field of transfer learning.

**Weaknesses:**

One weakness of the paper is the limited exploration of continuous groups, as the focus is primarily on finite groups. The authors acknowledge the need for further work to extend their methods to continuous groups but do not provide concrete solutions or insights in this regard. This limitation restricts the proposed algorithms' generalizability and applicability to real-world applications involving continuous transformations. Addressing this weakness by offering a more detailed discussion on approaches for handling continuous groups would enhance the paper's relevance and broaden its potential impact.

**Questions:**

(1) This paper mentioned that equizero performs well when good agent loss functions are available for downstream tasks. Can the authors elaborate on the process of selecting or designing these proxy loss functions? How did they ensure that the selected loss functions accurately capture the desired goals of each task?

(2) Could the authors discuss the potential implications of their research in real-world applications and any limitations or considerations when deploying the proposed methods in practical scenarios?

**Limitations:**

Yes

---

> ### Author Rebuttal · Authors · 2023-08-09
>
> We appreciate that the reviewer finds our approach novel, innovative, and applicable in multiple domains and models. We address the weaknesses and questions raised by the reviewer below:
>
> weaknesses:
>
> Reviewer: "...limited exploration of continuous groups..."
>
> **Response:** We thank the reviewer for pointing out the significance of exploring continuous groups to our setting.
> We have only considered discrete groups for our work since we were mainly motivated by our considered applications that only required discrete groups.
>
> To this end, we first point out that the work of Kim et. al. (that appeared on arxiv after the Neurips submission deadline), as pointed out by Reviewer DNsq, generalizes our framework to continuous groups using symmetric probabilistic averaging leading to equivariance in expectation for continuous groups.
>
> We would also like to note that extending our framework to continuous groups is simple and there could be many ways to generalize our framework to continuous groups.
>
> As such, we show that it also can be done by combining the setting of Kaba et. al. with our $\lambda$-equitune setup. **We have provided the extension with proof of equivariance in the response to Reviewer DNsq and the experimental results are in the pdf corresponding to the global response.**
>
> It shows that $\lambda$-equitune can be used to weigh the features corresponding to different transformations of the inputs obtained from canonicalization differently, leading to improved performance.
>
> We hope this convinces the reviewer that our work can be extended beyond discrete groups easily.
>
> Questions:
>
> Reviewer: "This paper mentioned that equizero performs well when good agent loss functions are available for downstream tasks. Can the authors elaborate on the process of selecting or designing these proxy loss functions? How did they ensure that the selected loss functions accurately capture the desired goals of each task?"
>
> **Response:** We thank the reviewer for raising this important question. We found that several machine learning tasks contain naturally available loss functions that can act as a proxy of their performance, e.g. CLIP has similarity scores, and Q-learning has Q-values. Maximizing these scores naturally leads to better performance as it is part of their original optimization tasks. The task of fairness in GPT2 is similar to that of the RL task, where we assign values describing the performance of the models. For GPT2 the scores are directly obtained from the regard scores of Sheng et. al. For compositional generalization in languages, the task is similar to classification and we chose the maximum of the probability as the score since it shows the confidence of the model in making the prediction (alternately one could also use the entropy of the probability vector, which is often used to measure confidence score of prediction). We found this score to work well for compositional generalization. The only surprising case that we found was that of classification using CNNs, where simple averaging of equitune seems to outperform loss /score functions such as entropy/maximum of probability, which shows that there are cases where finding good loss functions might be non-trivial. This, in addition to noting that the loss function can be replaced with learnable weights as score function leads to our general method, called lambda-equitune.
>
> Hence, we conclude that several machine learning tasks have naturally available loss/score functions that can be exploited for equizero. However, lambda-equitune can always be employed to obtain good equivariant finetuning results irrespective of the domain of application.
>
> Reviewer: "Could the authors discuss the potential implications of their research in real-world applications and any limitations or considerations when deploying the proposed methods in practical scenarios?"
>
> **Response:** We believe equizero and lambda-equitune when used with strong pretrained models would definitely make them robust, e.g. uses in robotics or object recognition.
> Regarding the fairness studies in our work, the current version focuses on debiasing using the metric and setup of Sheng et. al. and using the equality and neutral word sets from Basu et. al. that were human-made. We believe that for deployment of this application requires further testing and improvement in both the used regard scores and the construction of the equality and neutral sets. In particular, both the regard scores, and the equality and neutral word sets need to be constructed such that they satisfy any requirements of the application where it is deployed.
>
> Basu et al., Group Equivariant Fine-Tuning of Pretrained Models (2023)
>
> Kim et al., Learning Probabilistic Symmetrization for Architecture Agnostic Equivariance (2023)
>
> Kaba et al., Equivariance with Learned Canonicalization Functions (2022)
>
> Sheng, et. al. The woman worked as a babysitter: On biases in language generation. (EMNLP-IJCNLP), 2019.

---

> > ### Comment · Reviewer_8HWc · 2023-08-16
> >
> > I am highly satisfied with the authors' responses; their answers have completely addressed my concerns. I recommend accepting this paper in light of the other reviewers' comments.

---

### Official Review · Reviewer_vsSr · 2023-07-06

**Soundness:** 3 good
**Presentation:** 3 good
**Contribution:** 3 good
**Rating:** 6
**Confidence:** 3

**Summary:**

This paper proposed an equivariant few-shot learning method from pretrained models, namely λ-equitune, averaging the features using importance weights, i.e. λs. These weights are learned directly from the data using a small neural network, leading to excellent zero-shot and finetuned results that outperform equitune. This work further proves that λ-equitune is equivariant and a universal approximator of equivariant functions, and shows that equitune and equizero (the method of Kaba et al. (2022) used with appropriate loss functions) are special cases of λ-equitune. The authors conduct a series of analyses and experiments, validating the simplicity and generality of the proposed method on a wide range of diverse applications and models.

**Strengths:**

(1). The idea of this work is novel for equivariant few-shot learning from pretrained models.

(2). A wide range of diverse applications and experiments validate the claim of the work.


**Weaknesses:**

(1). I think Sub-Section '3.2 Properties' is somewhat confusing and should be analyzed in detail to ensure logical consistency. For instance, what is the connection between Theorem 1 and Definition 1, Theorem 2? Is Theorem 1 the foundation of Definition 1 and Theorem 2? Mathematical tools such as definitions, theorems, lemmas, etc. are used to summarize and abstract the theory of the entire method, requiring detailed descriptions to construct their connections.

(2). Using importance weights to the features averaging in λ-Equitune need to be discussed, including its benefits against to existing equivariant finetuning methods. Could learnable λ weights adapt the feature outputs from pretrained models? Is this a key to λ-Equitune obtaining excellent results in few-shot learning?


**Questions:**

In Equation (3), maybe M_G^λ (g,x) is M_G^λ (gx)?



**Limitations:**

Limitations are provided in the paper.

---

> ### Author Rebuttal · Authors · 2023-08-09
>
> We thank the reviewer for finding our work novel and appreciating the diversity of our experiments.
>
> We address the weaknesses and the questions raised by the reviewer below:
>
> Reviewer: "I think Sub-Section '3.2 Properties' is somewhat confusing and should be analyzed in detail to ensure logical consistency. For instance, what is the connection between Theorem 1 and Definition 1, Theorem 2? Is Theorem 1 the foundation of Definition 1 and Theorem 2? Mathematical tools such as definitions, theorems, lemmas, etc. are used to summarize and abstract the theory of the entire method, requiring detailed descriptions to construct their connections."
>
> **Response:** We thank the reviewer for pointing out the confusion in the explanation of the theoretical results in Sec. 3.2. We clarify the theoretical contributions below and will update it to the revised version.
>
> Theorem 1 is a standalone theorem showing that the proposed method called lambda equitune is provably equivariant to the considered group G.
>
> Definition 1 provides a definition of universality popularly used in several group equivariant neural networks papers, Yarotskiy (2022), Ravanbakhsh (2020). Theorem 2 uses the definition of universality from Definition 1 in its proof of showing that lambda equitune is a universal approximator of equivariant functions. The proof to Theorem 2 is provided in the appendix.
>
> We hope this improves the exposition of the results in Sec. 3.2.
>
>
> Reviewer: "Using importance weights to the features averaging in λ-Equitune need to be discussed, including its benefits against to existing equivariant finetuning methods. Could learnable λ weights adapt the feature outputs from pretrained models? Is this a key to λ-Equitune obtaining excellent results in few-shot learning?"
>
> **Response:** Indeed the idea behind lambda equitune is that the learnable lambda weights adapt such that they are high for the "good" features and low for "bad" features. Here "good" features are the features that contribute towards better performance and similarly, the "bad" features do not help get good performance. Existing equivariant finetuning methods such as equitune simply average all the obtained features from different transformed inputs, which is deleterious because not all features contribute equally to the performance of the model.
>
> The reason why certain transformations of the input yield better results can be explained through the example of using CLIP on transformed imagenet and CIFAR100 in Fig. 4a and 7, respectively. Note in both the figures that CLIP shows better results when the input images are provided in the upright form. Whereas, when the input images are rotated or flipped, their performances drop significantly, thus showing that the features obtained from non-upright images are not as useful aa the ones obtained from the upright images.
>
> Now, when a dataset contains images that have random rotations and flips, the lambda weights can automatically find out which transformed images contribute the most to the performance (in this case the upright images) and weight them the most leading to better performance than equitune of Basu et. al.
>
> We hope this explains the importance of the lambda weights better. We would add this to the revised version of the paper to help improve the clarity.
>
> Reviewer: "In Equation (3), maybe M_G^λ (g,x) is M_G^λ (gx)?"
>
> **Response:** We thank the reviewer for pointing out this typo. We will update this in the revised version of the paper.
>
> Ravanbakhsh. "Universal equivariant multilayer perceptrons." ICML 2020.
>
> Yarotsky. "Universal approximations of invariant maps by neural networks." Constructive Approximation (2022)

---

> > ### Comment · Reviewer_vsSr · 2023-08-17
> >
> > I appreciate the detailed response from the authors, which have sufficiently addressed my concerns. I am happy to accept this paper.

---

### Official Review · Reviewer_DNsq · 2023-07-07

**Soundness:** 3 good
**Presentation:** 3 good
**Contribution:** 3 good
**Rating:** 7
**Confidence:** 3

**Summary:**

This paper proposes an extension of symmetrization approach (Yarotsky 2018; Puny et al., 2021; Kaba et al., 2022; Basu et al., 2023) for achieving invariance and equivariance to (small and finite) symmetry groups, with a focus on empirical demonstration of zero- and few-shot transfer learning from non-equivariant pretrained architectures for a range of applications involving different group symmetries. The technical contribution that allows zero- and few-shot transfer is the introduction of some score (rank) function denoted lambda that weights all possible transformed inputs, and using it to turn standard group averaging (Yarotsky 2018) into weighted averaging in a way that the equivariance (and universality) of symmetrization is still guaranteed. The key idea is that an appropriate choice of the score function allows symmetrization to work favorably for the underlying pretrained model (by assigning higher weights to "important" group transformations), which can allow few-shot transfer learning. The authors name this approach lambda-equitune. In particular, choosing the score function as an indicator on argmin of some loss function allows for a canonicalizing symmetrization (Kaba et al,., 2022) that empirically allows zero-shot transfer, which the authors name equizero. The authors experimentally demonstrate the proposed algorithm in a range of applications including reinforcement learning, fairness in language models, compositional generalization, and image recognition under 90-degree rotations and flips.

Yarotsky et al., Universal approximations of invariant maps by neural networks (2018)

Puny et al., Frame Averaging for Invariant and Equivariant Network Design (2021)

Kaba et al., Equivariance with Learned Canonicalization Functions (2022)

Basu et al., Group Equivariant Fine-Tuning of Pretrained Models (2023)

**Strengths:**

S1. The paper aims to address an important and original problem of few-shot or zero-shot transfer of non-equivariant pre-trained deep neural networks to solve equivariant problems. While equivariant transfer learning from non-equivariant model has been investigated by some prior and concurrent work (Basu et al., 2023; Kim et al., 2023), few-shot or zero-shot transfer has not been demonstrated in literature as far as I know. The methodology is clearly motivated and explained, and I think this offers a nice way to steer the behavior of pretrained models towards equivariance with additional controllability offered by the score function or loss function, as demonstrated in the fairness experiment where fairness jointly with high regard score is achieved.

S2. The score function lambda for lamdba-equitune and the loss function for equizero doesn't seem to have to respect group symmetry, which is an advantage as it allows for wider range of choices. While this has been theoretically shown in Kaba et al., 2022, as far as I know this is the first work to empirically utilize the property, since Kaba et al., 2022 proposed but did not experiment with optimization approach.

S3. The applications demonstrated with experiments are quite comprehensive, ranging from reinforcement learning to language generation and visual recognition. Also, the experimental results overall seems to support the main claims of the paper.

Basu et al., Group Equivariant Fine-Tuning of Pretrained Models (2023)

Kim et al., Learning Probabilistic Symmetrization for Architecture Agnostic Equivariance (2023)

Kaba et al., Equivariance with Learned Canonicalization Functions (2022)

**Weaknesses:**

W1. One major ambiguity I find in the paper is that, while the title of the paper and some parts in the main text mention few-shot learning, it seems the experiments concern zero-shot learning or fine-tuning with a fair amount of data. This was confusing to me given that fine-tuning is not equivalent to few-shot learning. Am I missing something?

W2. The approach is only applicable to small, finite groups, due to the requirement of evaluating the score function (lambda) for all possible transformed inputs. This is in contrast to some concurrent work (Kim et al., 2023) that extends to combinatorial or continuous groups, and can be considered a limitation of the proposed algorithm at the current state.

W3. For the fairness in language generation, I think there is a limitation in the proposed algorithm that the considered words (upon which group transformations are defined) are implicitly assumed to be not separated by the tokenizer of the language model. This might not be generally true for modern language models, as a tokenizer can choose to split an equality word into non-equality substrings (a potential example is waitress -> wait + ress). In this case, fairness would not be achieved as expected using vocabulary permutation transformations.

Kim et al., Learning Probabilistic Symmetrization for Architecture Agnostic Equivariance (2023)

**Questions:**

Q1. For the experiments mentioned as few-shot (e.g., Figure 10), how many shots are used?

Q2. In case of zero-shot learning without parameter updates, what kind of practical advantage could we expect from the universality result (Theorem 2)?

**Limitations:**

The authors have addressed potential limitations and negative societal impact in Section 6.

---

> ### Author Rebuttal · Authors · 2023-08-09
>
> We appreciate that the reviewer finds our problem original and contributions advantageous.
>
> We first provide some clarifications, then address the weaknesses.
>
> Clarifications:
>
> Reviewer: "S2: The score  ... choices. While ... shown in Kaba et al., 2022, ... experiment with optimization approach."
>
> **Response:** To the best of our knowledge, Kaba et. al. only show that the loss function need not be equivariant in the case of the optimization approach where there is an arg min (cf. eqn. 4, eqn. 6 in Kaba et. al.). Instead, our setup simply performs weighted averaging. Of course, the setting of Kaba et. al. can be generalized to ours. However, this generalization is not shown in Kaba et. al.
>
> Weaknesses:
>
> Reviewer: "One major ambiguity ... title of the paper .... Am I missing something?"
>
> **Response:** We thank the reviewer for pointing out the confusion in title. We agree that the focus of the paper can be better described as a combination of zero-shot learning and finetuning rather than few-shot learning. We admit our number of training samples, even though much smaller than those used by pretrained models, should technically be called finetuning rather than few-shot learning. We used the umbrella term "few-shot learning" to explain the plethora of application across zero-shot learning and finetuning.
>
> We wanted to emphasize that while equituning takes several iterations on the data to obtain good results, equizero and lambda-equitune can perform much better than equitune with only a few iterations or none (equizero).
>
> We are happy to change the title to "Efficient Equivariant Transfer Learning from Pretrained Models", a better description for our contributions. We apologize for any caused confusion.
>
> Reviewer: "... applicable to small, finite groups... concurrent work (Kim et al., 2023)...."
>
> **Response:** Our framework can be easily extended to continuous groups by using it with canonicalization of Kaba et. al., which is an alternative to the method of Kim et. al. (*please note this work appeared only after the Neurips deadline*).
>
> **Main idea:** combine canonicalization from Kaba et. al. with $\lambda$-equitune leading to expressive equivariant network with weighted averaging over features with different group actions applied to them.
>
> **Def**: Given a (continuous) group $G$, a non-equivariant function $M:X\mapsto Y$, and equivariant auxiliary function (from the setting of Kaba et. al.) $h: X \mapsto G$, lambda functions $\lambda: X \mapsto R^+$, and a set of group elements $\Theta$ = {$\theta_1, \ldots, \theta_k$}, i.e. $\theta_i \in G$, we define the canonical-$\lambda$-equitune operators as
> $M^{ \lambda }_{G, equi}(x) = $
>
>  $(\sum_{ \theta \in \Theta } \lambda (\theta h(x)^{-1}x) h(x) M( \theta h(x)^{-1}x) )/( \sum_{ \theta \in \Theta }  \lambda (\theta h(x)^{-1} x) )$
>
>
> $M^{ \lambda }_{G, inv}(x) = $
>
>  $(\sum_{ \theta \in \Theta } \lambda (\theta h(x)^{-1}x) M( \theta h(x)^{-1}x) )/( \sum_{ \theta \in \Theta }  \lambda (\theta h(x)^{-1} x) )$
>
>
> **Thm**: $M^{\lambda}_{G, equi}(x)$ is equivariant to $G$.
>
> **Proof**: First note $h(gx) = g h(x)$.
> Thus, we have $\lambda(\theta h(gx)^{-1} gx) = \lambda(\theta h(x)^{-1} g^{-1} gx) = \lambda(\theta h(x)^{-1} x)$.
> Hence, $\lambda(\theta h(gx)^{-1} gx)$ is invariant to actions of $G$.
>
> Finally, $M^{ \lambda }_{G, equi}(g x)$
>
> $=(\sum_{ \theta \in \Theta } \lambda (\theta h(gx)^{-1}gx) h(gx) M( \theta h(gx)^{-1}gx) )/( \sum_{ \theta \in \Theta }  \lambda (\theta h(gx)^{-1} gx) )$
>
> $=(\sum_{ \theta \in \Theta } \lambda (\theta h(x)^{-1}g^{-1}gx) g h(x) M( \theta h(x)^{-1}g^{-1}gx) )/( \sum_{ \theta \in \Theta }  \lambda (\theta h(x)^{-1} g^{-1}gx) )$
>
> $=g (\sum_{ \theta \in \Theta } \lambda (\theta h(x)^{-1}x) h(x) M( \theta h(x)^{-1}x) )/( \sum_{ \theta \in \Theta }  \lambda (\theta h(x)^{-1} x) )$
>
> $=g M^{\lambda}_{G, equi}(x)$.
>
> The proof for invariance of $M^{\lambda}_{G, inv}(x)$ follows similarly.
>
> **Exp. results are provided in the Tab. global response pdf**
>
> For exp., we use $G = $SO(2), the invariant regression function from Finzi et. al. as our task. We define $M$ as an MLP with 5 layers. $h$ is constructed using a fixed function that sums $x_1, x_2$ and computes the corresponding SO(2) rotation matrix from it. $\lambda$ is a small MLP with 3 layers, but much smaller number of neurons in them. We adjust the num. of params. in models to make sure both when using $\lambda$ and without, we have similar number of parameters.
>
> We use a train and test size of 10000, 10000, batch size 500, learning rate $5*10^{-3}$, epochs 100, num. of seeds 5.
>
> Finzi et. al.. "A practical method for constructing equivariant multilayer perceptrons for arbitrary matrix groups." ICML, 2021.
>
> Reviewer: "For the fairness... limitation...tokenizer of the language model."
>
> **Response:** We agree with the reviewer that the tokens might not be words.
>
> As discussed in limitations in lines 318-319, we think there is a scope for optimizing these equality and neutral sets instead of using human-made sets of words.
>
> In future, one can directly learn sets of tokens that maximizes regard scores of LLMs, e.g., using RLHF.
>
> Note, our current formulation still give empirically good fairness results for GPT2 (cf. Fig. 3 and 9) using BPE tokenizers.
>
> Questions:
>
> Reviewer:"..., how many shots are used?"
>
> Please note, as discussed above, our motivation is to show efficiency of finetuning (using few iterations) in this Fig. We are happy to update the title and figure labels. Apologies for confusion.
>
> Reviewer:"In case of zero-shot learning without parameter updates, what kind of practical advantage could we expect from the universality result (Theorem 2)?"
>
> **Response:** It says that if the non-equivariant model is well-trained and is a good approximator to a certain function, then so are the obtained equivariant zero-shot results. This result ensures that obtained equivariant model is still an expressive equivariant model and not overconstraining the pretrained model.

---

> > ### Comment · Reviewer_DNsq · 2023-08-11
> > **Response to rebuttal**
> >
> > Thank you for the comprehensive response. Overall, I find that the rebuttal clearly addresses most of my concerns. I especially appreciate pointing out that score based canonicalization was only considering argmin while this paper extends to weighted average, and also the added extension to large groups, as well as added explanation on usefulness of universality on zero-shot learning. I also agree that revising the title to "Efficient Equivariant Transfer Learning from Pretrained Models" would resolve my concern on few-shot learning.
> >
> > For the review on continuous groups, I was not intending to make explicit comparison to Kim et al., (2023) -- as the authors pointed out, it has been on arXiv after NeurIPS deadline -- but the review was to point out the limitation of the proposed method on large groups. Now I can see that the issue has been resolved with the extended algorithm of weighted averaging combined with canonicalization. Furthermore, it has been also demonstrated empirically with a compelling performance.
> >
> > Overall, I am happy to raise my score from 5 to 7.

---

### Official Review · Reviewer_TNLv · 2023-07-12

**Soundness:** 3 good
**Presentation:** 3 good
**Contribution:** 2 fair
**Rating:** 5
**Confidence:** 4

**Summary:**

This paper proposes lambda-equitune based on previous works, which averages the features using learned weights by data and small neural network. This paper provides detailed theoretical analysis to prove that the proposed lambda-equitune is equivariant and a universal approximator of equivariant functions. Diverse experiments are conducted to show the generality of the method.

**Strengths:**

1. Efficient transfer learning from foundation models to downstream tasks is an important tasks. This work make interesting improvements on previous work by introducing important weights trained from data and small neural network.
2. Diverse experiments results are provided, including image classification, deep Q-learning and natural language generation. The proposed lambda-equitune method shows good results on majority of the tasks.
3. Detailed theoretical analysis for equitune is provided. The writing and illustration is clear.


**Weaknesses:**

1. The novelty against the previous work of Basu et al. (2023). Since extra parameters and fintuning process are introduced, the contribution of this work could be further explained.
2. According to figure 1, both original data and transformed data are used for inference of the proposed method, it is very natural to conduct embedding from features or from results, how about these embedding methods compared with the proposed method?
3. The setting of the image classification experiments is a bit naive. Since the paper focuses on efficient transfer learning, there are many more meaningful and realistic transfer learning tasks in computer vision domain rather than flip or rotation of 90 degree.
4. Minors: the format of reference part should be adjusted in Page 12.


**Questions:**

Pease refer to the weaknesses part.

**Limitations:**

Yes

---

> ### Author Rebuttal · Authors · 2023-08-09
>
> We thank the reviewer for finding our contribution interesting.
>
> We first clarify a few points from our paper that the reviewer might have misunderstood, then we address the weaknesses pointed out by the reviewer.
>
> Reviewer: ``This paper proposes lambda-equitune based on previous works, which averages the features using learned weights by data and small neural network."
>
> **Response:** Please note that we not only provide the general framework of \lambda-equitune but also provide an important special case called equizero that does not require any additional neural network or even any additional data.
>
> We would like to point out that using equizero with appropriate loss functions (often naturally available, e.g. image-text similarity scores in CLIP) we outperform equitune in several experiments such as
> a) improving the robustness of CLIP b) deep Q-learning c) Debiasing and detoxifying (improving the regard scores) LLMs, using regard scores as the loss/score function d) improving compositional generalization capabilities of RNNs, GRUs, and LSTMs
>
> We use $\lambda$-equitune in cases where searching for a loss function is non-trivial. We illustrate one such case in the paper: classification with CNNs.
>
> Now, we address the weaknesses pointed out by the reviewer.
>
>
> Reviewer: ``The novelty against the previous work of Basu et al. (2023). Since extra parameters and fintuning process are introduced, the contribution of this work could be further explained."
>
> **Response:** There are two main novelties of this work that benefit our work compared to Basu et. al:
> a) equizero: equivariant zero-shot learning using no additional data or small neural networks. This outperforms equitune for zeroshot learning on several diverse downstream tasks.
> b) \lambda-equitune: where we use a small neural network to weigh the features obtained from the pre-trained model corresponding to different transformed inputs.
>
> From Tab. R1, note that the number of added trainable parameters is negligible. It shows that using a tiny fraction of extra parameters for performing the weighted averaging can be highly beneficial for extracting equivariant features from pretrained models.
>
> Table R1. Number of additional trainable
> | Exp. name       | added trainable params| pretrained params | frac of added params|
> | ----------- | ----------- |----------- | ----------- |
> CLIP (RN50)		|	112.5k	     	|    25.6 M               |	0.0043	     |
> CLIP (RN101) 		|	61.3k		|    44.5 M	|	0.0013	|
> CLIP (ViT-B/32 and ViT-B/16)|	61.3k	 	|    86M		|	0.0007	|
> Resnet			|	66.6k		|    11.6M	|              0.005	|
> Alexnet			|	934.k		|     61.1M	|              0.0150		|
>
>
>
> Reviewer: "According to figure 1, both original data and transformed data are used for inference of the proposed method, it is very natural to conduct embedding from features or from results, how about these embedding methods compared with the proposed method?"
>
> **Response:**
> Unfortunately, since the reviewer has not provided any reference for the embedding methods they are referring to, we are unable to provide a detailed comparison with these methods. We would urge the reviewer to kindly share some references so that we could try to provide some comparison.
>
> Please note that several of our experiments use embedding from the CLIP model and use image-text similarity scores to obtain the classification scores.
>
> In Fig. 4a and 7, we find that existing CLIP embedding-based classifiers that do not use group equivariance are not robust to transformations such as flip or rotation of 90 degrees. Then, in Fig. 4b, 4c, and 8, we show how equizero leverage group equivariance to provide robustness to such transformations, moreover, it outperforms other equivariant methods such as equitune without using any additional data or learnable parameters. Experiments on lambda equitune that use CLIP-based embedding techniques are provided in Fig 5, 11.
>
> Moreover, since our formulation of both lambda-equitune and equizero are model agnostic, it is easy to extend them to other embedding based methods not considered in this work, such as in [1]. In [1], we can simply make both the CLIP and the cache models equivariant to guarantee equivariance of this embedding based technique.
>
> We believe this convinces the reviewer that a) we have already provided comparisons with CLIP-based embedding methods, b) the generality of our method allows to incorporate our method into any other embedding based methods.
>
> [1] Zhang et. al. "Tip-Adapter: Training-free Adaption of CLIP for Few-shot Classification", arXiv:2207.09519v1
>
> Reviewer: "The setting of the image classification experiments is a bit naive. Since the paper focuses on efficient transfer learning, there are many more meaningful and realistic transfer learning tasks in computer vision domain rather than flip or rotation of 90 degree."
>
> **Response:**
> Note our method is completely general and provably equivariant for any group. Our experiments focusing on robustness to flips/rotations for pretrained CLIP/CNN models can be easily generalized to any other discrete groups, when applicable.
>
> We emphasize that our work is not restricted to computer vision. Besides, we also show efficient transfer learning in several other domains such as Fairness in NLG (e.g., debiasing GPT2), deep Q-learning, compositional generalization in languages.
>
> Moreover, recent work [2] shows that equivariance to seemingly naive groups such as rot90/flip can provide improvement in performance where the actual transformations in data are much more complicated and not explicitly known.
>
> [2] Wang et. al. "The Surprising Effectiveness of Equivariant Models in Domains with Latent Symmetry", ICLR 2023
>
> Reviewer: "Minors: the format of reference part should be adjusted in Page 12."
>
> **Response**: We thank the reviewer for pointing this out. We will make sure to fix this in the updated version.

---

> > ### Comment · Reviewer_TNLv · 2023-08-18
> >
> > Most of my concerns are addressed by author's response. I tend to accept this paper.

---

### Author Rebuttal · Authors · 2023-08-09

We sincerely thank the reviewers for their valuable comments and suggestions regarding our paper titled, “Equivariant Few-Shot Learning from Pretrained Models”. Our paper is motivated by the need to efficiently utilize pretrained models for a variety of downstream applications which benefit from equivariance. To that end, we propose equizero, a zero-shot model that is equivariant and with much better performance than equituning. This is because equizero chooses the best features from pretrained models using a proxy loss function, unlike equitune that simply performs averaging over features. We also show important theoretical properties of this equivariant model like universality, which says that our method still preserves the ability to approximate any equivariant function. We further demonstrate diverse downstream applications of equi-zero over wide-varying tasks like image classification, reinforcement learning, compositional generalization and equivariant CLIP. We also propose $\lambda$-equitune, which is a relaxed generalization of the equizero algorithm and does not require any proxy loss whatsoever. Experiments validate the efficiency of $\lambda$-equitune and equizero over equitune. We next try to alleviate a few key concerns pointed out by reviewers

**Key Concerns**

**Approach limited to finite groups:**

We address the concerns raised by reviewers DNsq and 8HWc regarding the limitations of our approach to finite groups.

Reviewer DNsq also refers [3], which extends our framework to continuous groups by performing symmetric probabilistic averaging.

First, we gently point out that [3] **appeared online only after the Neurips deadline**. Further, we would like to show there are alternate simple methods to extend $\lambda$-equitune to continuous groups. We describe one such method here.

We simply use the canonicalization method of [1] in conjunction with $\lambda$-equitune to obtain $\textit{canonical}$-$\lambda$-$\textit{equitune}$, which is a) equivariant to continuous groups and b) weighs different features using importance weights to obtain expressive equivariant network.

This idea is based on constructing equivariant frames of [2]. [1] uses a frame of size exactly one, whereas we use a frame of larger size and weigh the features corresponding to different frame elements based on their importance.

A formal definition of our method and its proof of equivariance is provided in the response to reviewer DNsq.

Further, we conduct an experiment on the SO(2) invariant regression task of [4, Sec. 7. 1]. The results of this experiment are provided in the attached table. It shows that canon-$\lambda$-equitune clearly outperforms non-equivariant model as well as canonicalization method of [1].

Thus, a simple extension of $\lambda$-equitune leads to equivariance to continuous groups.

[1]: Kaba et. al., “Equivariance with Learned Canonicalization Functions”, ICML 2023

[2]: Puny et. al., “Frame Averaging for Invariant and Equivariant Network Design” ICLR 2022

[3]: Kim et. al., “Learning Probabilistic Symmetrization for Architecture Agnostic Equivariance”, ArXiv 2023

[4]: Finzi et. al. "A practical method for constructing equivariant multilayer perceptrons for arbitrary matrix groups." ICML. 2021.

**Ambiguity about title**

We agree that the focus of the paper can be better described as a combination of zero-shot learning and finetuning rather than few-shot learning. It should be technically called finetuning rather than few-shot learning. We used the umbrella term "few-shot learning" to explain the plethora of application across zero-shot learning and finetuning. However, we admit that changing the title to a better name would be appropriate.

As such, we are happy to change the title of our paper to "Efficient Equivariant Transfer Learning from Pretrained Models", which would be a better title to describe the combination of zeroshot and finetuning. We apologize for any caused confusion and thank the reviewer for pointing this out. We believe this will help readers to understand our work better.

**Image classification experiment for $\lambda$-equitune is a bit naive**

First, we would again emphasize that our method is completely general and provably equivariant for any transformation for which the group actions on the input/output are well-defined. Thus, our experiments focusing on robustness to flips/rotations for pretrained CLIP/CNN models can be easily generalized to any other discrete groups.

Secondly, we emphasize that our work is not restricted to computer vision. Apart from experiments on computer vision (CLIP and CNN-based equivariant/robust classification), our work also shows efficient transfer learning in several other domains such as Fairness in natural language generation (e.g., debiasing and detoxifying GPT2), equivariant deep Q-learning, compositional generalization in languages.

Moreover, recent work [5] shows that equivariance to seemingly naive transformations such as rot90/flip can provide robustness/improvement in performance where the actual transformations in data are much more complicated and not explicitly known. This shows that our experiments with equivariance to seemingly simple transformations can be useful in much more complicated scenarios and could motivate further investigation in this direction in future works.

[5]: D.Wang et. al., The Surprising Effectiveness of Equivariant Models in domains with Latent Symmetry, ICLR 2023

---

### Decision · Program_Chairs · 2023-09-21

**Decision:**

Accept (poster)

**Comment:**

The submission looks into the problem of turning a pre-trained and non-equivariant network into an equivariant network for a downstream equivariant task. It introduces an approach called $\lambda$-equitune which performs a weighted averaging of transformed input embeddings. The approach generalizes equivariant finetuning and optimization-based canonical representation, and special cases of $\lambda$-equitune are shown to be equivalent to the two. Empirical results are presented on a range of applications including reinforcement learning, group-theoretic fairness in language, zero-shot compositional generalization, and image classification.

Reviewers note that the submission tackles an important task (TNLv, DNsq)–equivariant transfer learning in zero-shot and finetuned settings–presents a detailed theoretical analysis of the proposed approach (TNLv, 8HWc), and provides empirical support showing promising results on a diverse set of tasks (TNLv, DNsq, vsSr, 8HWc).

The most salient concerns raised by reviewers are:

* The proposed approach's novelty in the context of Basu et al. (2023) (TNLv). This concern is addressed by the authors' clarification of their contributions in the rebuttal.
* The image classification setup appears naive (TNLv). The authors emphasize in their response that their method is "completely general and provably equivariant for any transformation for which the group actions on the input/output are well-defined", and that the experiments can be generalized to any other discrete group, when applicable. This addresses Reviewer TNLv's concern.
* The term "few-shot learning" in the title and text is not reflective of the experiments carried out in the paper (DNsq). The authors acknowledge the concern in their response and offer to change the paper's title and wording to address it.
* The approach is only applicable to small, finite groups (8HWc, DNsq). In their response to Reviewer DNsq, the authors outline a strategy for extending their framework to continuous groups through the canonicalization of Kaba et al. (2022) and provide empirical results on this strategy. This addresses the reviewers' concerns.

All reviewers note in their replies to the authors that they are in favor of acceptance.